



# Comprehensive inventory of large hydropower systems in the Italian Alpine Region

Andrea Galletti[1,2], Soroush Zarghami Dastjerdi[2,3], and Bruno Majone[2]

[1]EURAC Research, Center for Climate Change and Transformation, Bolzano, Italy
[2]University of Trento, DICAM, Trento, Italy
[3]University School for Advanced Studies IUSS Pavia, Pavia, Italy

**Correspondence:** Andrea Galletti (and.galletti92@gmail.com)

**Abstract.**

Climate change raises the critical need to understand its impact on water resources, particularly as hydropower's role as a flexible, renewable energy source becomes more vital in planning for the energy system's decarbonization. While hydrological modeling represents an established tool for assessing the future evolution of water resources, a key challenge lies in its reliance on data describing the geometry and operation of hydropower systems interacting with the natural stream network. The Italian Alpine Region (IAR) is home to over 300 large hydropower systems (LHS), and its hydrological cycle is expected to suffer major alterations due to climate change. However, detailed and reliable hydrological studies in this region face hindrances due to the absence of a consistent, comprehensive, and openly available LHS source.

We present IAR-HP (Italian Alpine Region HydroPower), a comprehensive inventory specifically designed for the inclusion in hydrological modeling of LHS located in the Italian Alpine Region, to overcome this obstacle. This dataset aims to support modelers in the water-energy nexus by providing crucial information for accurately informing their models. Compiled from various online sources, IAR-HP is openly accessible and reproducible, offering a solution to the scarcity of data hindering effective storage hydropower-related simulations. The dataset was validated through a hydropower production modeling exercise, and was able to reconstruct 96.2% of the observed hydropower production across the Italian Alpine Region. By presenting this dataset, we contribute a practical tool for scientists to reduce the inherent uncertainty of hydrological models, improving their ability to represent large hydropower systems accurately. IAR-HP holds potential for numerous applications to inform decision-making in the dynamic context of climate change.

## 1 Introduction

Hydropower stands as a cornerstone of Italy's renewable energy context, harnessing the power of its rivers and mountainous terrain, and has consistently contributed around 15-20 % to Italy's total electricity generation since the 1980's (Ember, 2024), with more than 80% of it being produced within river catchments originated in the Alps (Terna, 2024), which are home to more than 200 large dams, with the main or sole purpose of hydropower production, developed from late 1800s until 1980s (MIT). Notwithstanding the increasing integration of solar and wind renewable power started in the early 2010s Ember (2024)), and without significantly increasing its installed capacity since the late 1980s, hydropower has remained vital due to its power





dispatchment and storage capabilities, that make it a reliable backbone in an increasingly renewable energy mix that leverages a growing share of intermittent power sources.

    Hydropower systems in the Alpine Region are expected to face significant impacts due to climate change, primarily from alterations in precipitation patterns, snowmelt timing (Gaudard et al., 2013; Maran et al., 2014; Gaudard et al., 2014), and the frequency of extreme weather events. In the Italian Alps, these changes could lead to significant changes in the seasonality of

water inflows to storage reservoirs, thus affecting the predictability and reliability of hydropower generation (Majone et al., 2016). A continuous reduction of Alpine glaciers is projected to affect hydropower production negatively (Patro et al., 2018), while earlier snowmelt caused by rising temperatures is expected to increase spring runoff (Wagner et al., 2016), while at the same time granting lower water availability during the critical summer months when the conflicts among different water uses are more pronounced (Maran et al., 2014; La Jeunesse et al., 2016). Indeed, several sectors throughout the Alps rely on timely

water availability: agriculture, hydropower production, industrial cooling, and drinking water supply are the most prominent examples. Integrated modeling perspective have become a customary approach among hydrologists (Falloon and Betts, 2010), to cope with the complex interaction among different sectors and come up with sound adaptation strategies in face of the hydro-climatic uncertainty anticipated by climate change (Howells et al., 2013). The WEFE (Water Energy Food and Ecosystems) nexus has catalyzed a wealth of attention: a clear majority of the research conducted over Mediterranean case studies involves

assessments regarding the water-energy side of the nexus, followed by the water-energy-food triangle (Lucca et al., 2023), highlighting the societal importance of these sectors and their intertwined dynamics. In light of these considerations, it is clear that any underlying assumption will have cascading effects on the results of any modeling endeavor. Concerning modeling the water-energy nexus over large-scale domains, the largest share of uncertainty can be attributed to two key factors: i) characterization of the existing hydraulic infrastructure and ii) their management by the operating companies. As per the latter,

it is no secret that actual management strategies (for the reservoir- and pumped-storage hydropower) are kept confidential by companies (Schaefli, 2015). Hence this variable is often reconstructed based on different proxies, or assumed resorting to generalised approaches (see e.g., Finger et al., 2012; Shrestha et al., 2014; Fatichi et al., 2015; Turner et al., 2017; Galletti et al., 2021; Vu et al., 2023). The characterization of existing hydraulic infrastructures has originated many notable endeavors: GRanDD (Lehner et al., 2011), GOODD (Mulligan et al., 2020), and GDAT (Tianbo Zhang and Gu, 2023) are among the

most valuable global scale geo-referenced datasets of large dams, and contain information such as dam location, geometric and hydraulic properties. These datasets contain plentiful information for global scale assessments; however, they might not fully suit the goal of WEFE nexus assessments, as these often require detailed information on the location, timing, and amount of water diversions. With particular reference to the case study of the Italian Alpine Region, global-scale dam data sets encounter some limitations: firstly, run-of-river plants (which by definition are not connected to large storage dams) are not included in

these datasets; secondly, large hydropower systems (LHS, by Italian regulation, those plants with an installed capacity greater than 3 MW) can be fed by relatively small dams with high hydraulic heads, once again resulting uncharted within global large dam datasets; thirdly, knowing the location and hydraulic properties of additional water intakes is of utmost importance to correctly model the spatial availability of water resources, as well as the river discontinuity caused by hydropower-related water abstractions. Data concerning hydropower plants over Europe are available through the Energy and Industry Geography Lab





portal (https://energy-industry-geolab.jrc.ec.europa.eu/), mostly solving the issue of "invisible" run-of-river plants mentioned earlier. However, the dataset does not record the location of their water intakes.

Motivated by the need for a comprehensive dataset for modeling hydropower (HP) production and its hydrological/environmental implications over the Italian Alpine Region (IAR), we propose IAR-HP, a dataset specifically aimed at providing a comprehensive anagraphic of the existing large hydropower systems, complete with as much geometric, hydraulic and man-

agement information as possible, to enable thorough assessments over such a complex and relevant domain. The dataset was compiled in order to comply with the information requirements (geometric, geospatial, and operational attributes of all LHS-related infrastructures) of the modeling approach described in Galletti et al. (2021), but its content can be easily adapted to any modeling framework. In the absence of official open-source information, we also provide validation of IAR-HP in terms of hydropower production modeling, conducting a modeling exercise over the entire IAR domain with the HYPERstreamHS

hydrological model (Avesani et al., 2021). The paper is structured as follows: firstly, we go through a comprehensive description of IAR-HP, starting from the definitions and assumptions that were made to complete the data collection and concluding with the presentation and discussion of some of the key statistics concerning the information gathered in IAR-HP. Secondly, we present a hydrologically-based exercise of hydropower production modeling that we conducted to validate the dataset. We close the paper with some remarks on the strengths and limitations of our dataset, hoping to boost its accessibility and

interoperability in other water-energy nexus assessments.

## 2 Methods

This section is divided into two main parts. The first part outlines the data collection procedure and details how the information was sorted and filtered to characterize large hydropower systems across the Italian Alpine Region. This includes a comprehensive explanation of the dataset framework structure, specifying the data types collected, the sources from which the data were

obtained, and the criteria for sorting and filtering the information. It further explains how these datasets interact with each other to provide a coherent and detailed characterization of the hydropower systems.

The second part focuses on the validation of the resulting dataset using a hydrological model. This exercise involves two main steps: firstly, we calibrate the model's hydrological kernel in several representative watersheds; secondly, we execute the model and compute time series of daily hydropower production at each LHS through dedicated routines. The validation

objective is to reproduce historical provincial hydropower production monthly time series within the IAR domain.

### 2.1 Database description

#### 2.1.1 Database design

According to the data reported by the Italian electrical grid manager (TERNA, Terna, 2024), more than 90% of the energy produced in IAR is derived from large hydropower systems. Based on Italian legal standards, hydropower systems are classified

as "small" or "large," with the threshold for large hydropower systems set at 3 MW installed capacity. Therefore, the IAR-HP





dataset focuses specifically on hydropower facilities with an installed capacity exceeding 3MW. This dataset provides comprehensive information on key factors such as identification characteristics, structural connections, and operational constraints for each large hydropower systems.

IAR-HP design and definition build on its initial implementation for the HYPERstreamHS hydrological model (Avesani et al., 2021), applied to the Adige catchment (Galletti et al., 2021), the third largest catchment in Italy, consisting of several complex hydroelectric systems. Following the satisfactory outcomes of this modeling endeavour, we attempted to expand IAR-HP following the same approach for the whole IAR domain to fill up the lack of a homogeneous, comprehensive, and freely accessible source of information concerning large hydropower systems.

The simulation of hydropower systems in HYPERstreamHS follows a nodes-links framework. In this framework, all hydropower-related infrastructures are represented as nodes, allowing them to be grouped based on their type: power plants, reservoirs, and water intakes. Each node type applies different constraints to water mass balance, reflecting the operational behavior of each structure, as extensively detailed in Avesani et al. (2021); Galletti et al. (2021).

### 2.1.2 Database collection

As IAR-HP was initially designed for large-scale hydroelectric facilities modeling, this approach embeds the phenomenal challenge of achieving a consistent level of detail throughout the database. These challenges involve gathering information from extremely inhomogeneous data sources such as regional or provincial webGIS, catchment authority reports, leaflets, construction plans, and web news, to name the most common. Table 1 summarizes the key geolocation, topologic, and qualitative attributes common to each node in IAR-HP, while the type-specific characteristics recorded for each infrastructure are summarized in 2. In the upcoming paragraphs, we will discuss how data collection issues were tackled and how the key features of different infrastructures were harmonized into IAR-HP.

X and Y coordination attributes of hydropower systems are the basic needs for providing a baseline of where these systems interact with water bodies. To this end, the main references were regional/provincial webGIS's and construction architectural plans, with the former providing most of the geolocation of all systems and the latter completing the information about the topological layout (i.e., connections).

Most of the topological information is stored in the node ID's. Each node is provided with an unique numeric identifier, $id_{NODE}$. Connections are achieved by referring to upstream and/or downstream node IDs ($id_{UP}$ and $id_{DOWN}$). In general, one node can have any amount of upstream nodes and only one downstream node; this was chosen because, in the rare occurrence where flow is diverted to more than one source, this involves complex and subjective maneuvering, a level of detail which is impossible to reliably reconstruct at the scale of IAR-HP. Finally, each node is assigned a type (Reservoir, Intake, or Plant) and three pertinence attributes (Province, Region, Basin) for easier filtering or analytical purposes.

The geo-topologic information was, however, not up-to-date or at all present for many regions: the initial analysis was refined by visually investigating the location of all systems through third-party resources such as OpenStreetMaps and Google Earth.

The z-coordinate plays a key role in our dataset, as it single-handedly defines the hydraulic head available to each LHS, directly correlating to its ability to produce energy. As a preliminary step, the z-coordinate was inferred by matching the x-y





**Table 1.** IAR-HP geolocation and topological characteristics for each node.

| Characteristic | Description | Data type [units] |
|---|---|---|
| $x_{cor}, y_{cor}$ | Location | CRS: WGS 84-UTM 32N [m] |
| $z_{cor}$ | Altitude | [m a.s.l.] |
| *Province* AND *Region* AND *Basin* | Pertinences | Node attribute |
| $P$ OR $I$ OR $R$ | Node type | Node attribute |
| $id_{NODE}$ | Node ID | Node attribute |
| $id_{UP}$ | Upstream node ID | array(Node attribute) |
| $id_{DOWN}$ | Downstream node ID | Node attribute |

coordinates and a high-resolution digital elevation model (DEMs) with a horizontal resolution of 30 meters. This first estimation was then refined, comparing the nominal head for all LHS with the head difference originated by the nodes in question. In case of minimal differences, the initial estimation was not modified for consistency. In contrast, in case of larger differences, the relevant z-coordinates were modified to match the nominal head declared for the LHS. This operation was mostly necessary in low-head, high-flow run-of-the-river plants, where a few meters can significantly impact the overall production. The resulting

x-y location of reservoirs and hydropower plants present in IAR-HP, as well as the diversion channels and penstocks connecting them, are displayed in Fig. 1. In the same Figure, the inset highlights the level of detail generally captured in IAR-HP with a specific focus on the Aosta province: this province hosts the largest number of RoR systems, and exhibits a complex network of diversion channels, which accuracy is one of the most valuable features of IAR-HP.

The specific characteristics gathered in IAR-HP for different node types are listed in Table 2. Hydropower plant entries

(Type-plant) represent plants with an installed power larger than 3MW. Plants are differentiated into two main operational categories: storage hydropower (R) and run-of-the-river (RoR) systems. Plants are further characterized by whether they involve either pumped- (PSH)or mixed-pumped storage hydropower(M-PSH); the difference between these two typings lies in how often the turbined water is pumped back to the upstream reservoir: if this happens on an almost daily basis, the plant is labeled as PSH, otherwise it is labeled M-PSH. Each plant is characterized by its installed power, $W_{inst}$ and a reference node

($id_{ref}$): reference nodes represent the node governing the water inflow to the plant, which is usually a reservoir in the case of reservoir hydropower systems, and the loading chamber (modeled as an intake) in the case of run-of-the-river systems. The gross hydraulic head is also computed as a difference between the two z-coordinates of plant and reference nodes. Penstock capacity information is likewise stored at the reference nodes for modeling purposes.



**Figure 1.** Distribution map of hydropower plants identified in the IAR-HP database, including relevant reservoirs for storage-based plants and diversion channels linking these systems. The top-left inset shows the Aosta province, highlighting the level of detail captured in IAR-HP. The bottom-left inset shows the covered area of IAR-HP in Italy.

Intake entries represent points at which water abstraction from the natural network is performed by means of different structures, varying from pipelines to channels and tunnels, that serve various purposes, such as restitution and diversion. Moreover, as detailed in Avesani et al. (2021), these nodes are also used to model confluences between two or more channels, where these only "merge" without accepting further external water abstraction.

Characterizing intakes embeds two significant challenges. First, technical plans often declare only the capacity of the final segment of the intake system, representing the maximum capacity at that point. This might, however, not be representative of the actual maximum abstraction capacity at the initial (more upstream) intake points, where pipe dimensions are likely smaller. Furthermore, when two pipes merge into a single final segment, the aggregated capacity does not clarify the relative contributions of each individual intake. Second, the intake installation point might not exactly match its corresponding water



**Table 2.** IAR-HP characteristics for each node type, adapted from Galletti et al. (2021).

| Characteristic | Description | Data type [units] |
|---|---|---|
| **Type-plant** | | |
| $R$ OR $RoR$ | Sub-type | Node attribute |
| $PSH$ OR $M\text{-}PSH$ OR $none$ | Pumping | Node attribute |
| $id_{ref}$ | Plant reference | Node attribute |
| $W_{inst}$ | Installed power | [MW] |
| **Type-intake** | | |
| $Q_{MAX}$ | Intake capacity | $[m^3 s^{-1}]$ |
| Confluence | Node Attribute | Logical [y/n] |
| **Type-reservoir** | | |
| $Q_{spill}$ | Spillway capacity | $[m^3 s^{-1}]$ |
| $Q_{Work}$ | Hydraulic capacity | $[m^3 s^{-1}]$ |
| $Q_{AVG}$ | Long-term average turbined flow | monthly$[m^3 s^{-1}]$ |
| $Q_{MEF}(t)$ | Minimum ecological flow | monthly$[m^3 s^{-1}]$ |
| Head | Gross system head | m |
| $Q_{RULE}(t)$ | Average turbine rate | $[m^3 s^{-1}]$ |
| V(H) | Stage-Storage curve | array[m a.s.l vs. Mm$^3$] |
| $H_{min,reg}$ | Minimum regulation stage | [m a.s.l.] |
| $H_{max,reg}$ | Maximum regulation stage | [m a.s.l.] |
| $H_{max,inv}$ | Spillway crest elevation | [m a.s.l.] |

body when compared to the river network extracted by the DEM (this second issue caused inconsistencies during modeling, with ambiguous flow attribution to intakes).

To address these characteristic deficiencies, we aggregate these intakes to the first location where the total channel capacity and the abstracted water body are known, maintaining their total pipe capacity to ensure consistent detail across the dataset:





this operation was performed in an attempt to preserve the maximum level of detail (i.e., not aggregating intakes if possible) while achieving unambiguous definition of both abstraction capacity and abstracted water body. The IAR-HP dataset provides attributes concerning the maximum water capacity at each intake point $Q_{MAX}$. If the intakes serve as a reference node for a

downstream plant, $Q_{MAX}$ represents that plant's hydraulic capacity. Intakes are also characterized by the Minimum Environmental Flow constraint, $Q_{MEF}(t)$, defining the minimum amount of water that must be present in the body before water can be abstracted.

Reservoir nodes are where water storage occurs, generally due to an impoundment allowed by a dam or natural lake. Each reservoir has its detailed operational regulations respecting three volume pool zones: flood control zone, active volume zone,

and inactive volume zone. Each dam characteristic contains zonal elevation thresholds that mark the boundary of each zone in the reservoir ($H_{max,inv}, H_{max,reg}, H_{min,reg}$). The structure of each reservoir is further characterised by the maximum discharge capacity of penstocks $Q_{Work}$ and spillways $Q_{spill}$. Additionally, stage-storage curves (V(H)) are provided for each reservoir, characterising its shape. Finally, each reservoir is constrained by the Minimum Environmental Flow constraint, $Q_{MEF}(t)$, and provided with a turbine discharge rate $Q_{RULE}(t)$, which development is extensively covered in Galletti et al.

(2021). This discharge rate follows a monthly fluctuation pattern, inferred from provincial production time series, capturing seasonal variations in hydropower utilization. This pattern is then applied to scale an average discharge rate for each reservoir, calculated based on the long-term mean production values declared for each corresponding hydropower system. This approach allows for a realistic, seasonally adjusted representation of turbine discharge rates across reservoirs, aligning modeled outputs with observed regional production dynamics with the overall long-term production of each reservoir hydropower system.

For both reservoir- and intake-type nodes, $Q_{MEF}(t)$ is defined according to the current legislation requirements (https://pianoacque.adbpo.it/deflusso-ecologico/, in italian): while the full legislation is more articulated and has not yet been uptaken by all provincial administrations, a simplified estimation of this requirement is often adopted, fixing it at 10% of the long-term average flow of each month. Likewise, we defined this quantity at each abstraction node by executing HYPERstreamHS without any hydropower modeling (natural conditions) and computing the 10% long-term average flows, achieving a consistent

definition of $Q_{MEF}(t)$.

### 2.1.3 Database statistics

The information gathered and the extensive dataset for large hydropower systems customized for the IAR resulted in an inventory covering twenty-five northern Italian provinces.

The IAR-HP database provides detailed and comprehensive information on 338 hydropower facilities, comprising 129 stor-

age and 209 run-of-the-river hydropower systems, totalling an installed power of 14.3 GW. Fig. 2 depicts the number of the LHS in the available provinces of IAR; in particular, the size of each pie chart is proportional to the total installed power in each province, while each pie itself represents the count of reservoir and run-of-the-river systems in each province. The distribution of LHS highlights a notable concentration in certain provinces, particularly Sondrio, Aosta, and Torino tied with Verbano-Cusio-Ossola, with 38, 35, and 32 LHSs installed, respectively. Sondrio also leads in terms of installed capacity across

the entire IAR, followed by Brescia and Trento. Additionally, Trento stands out with the highest number of storage hydropower



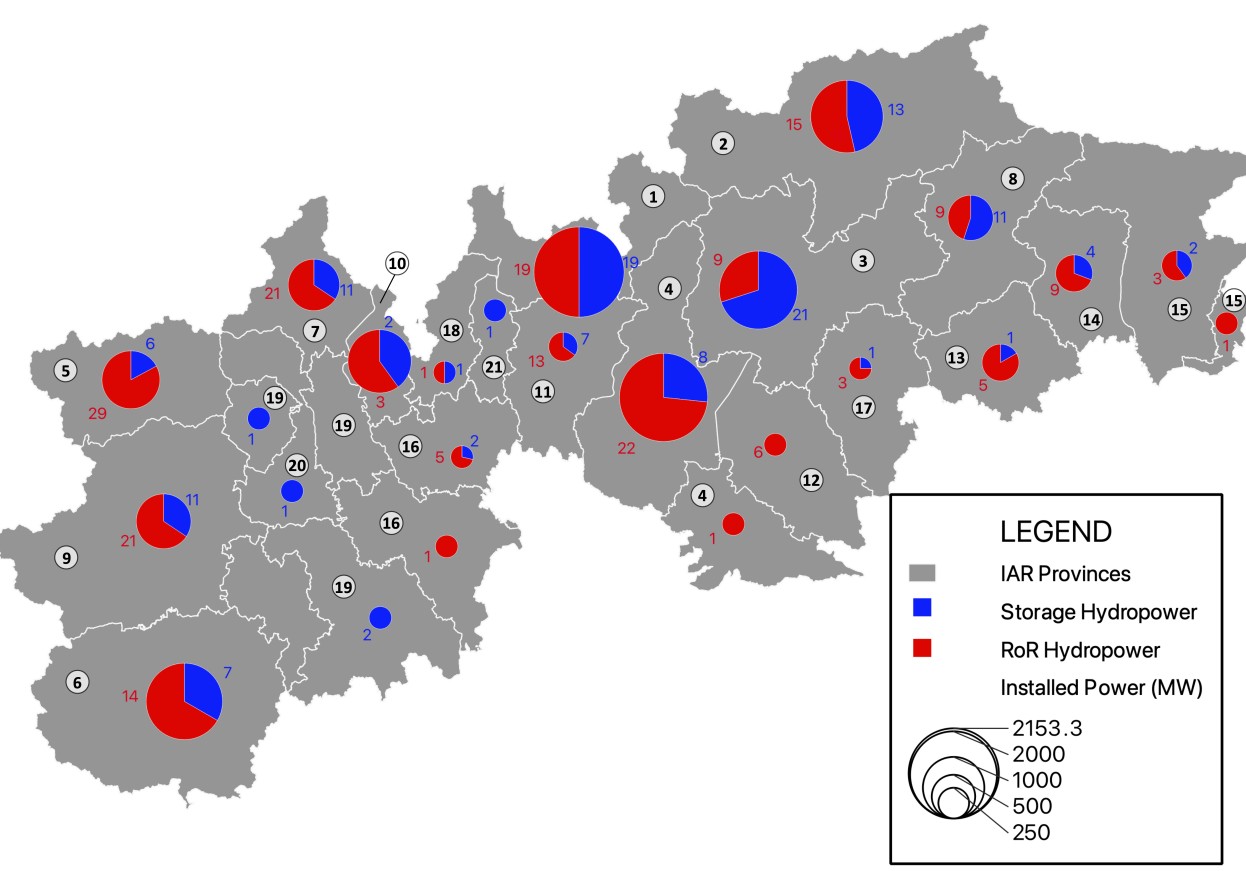

**Figure 2.** Statistic of the hydropower plants present in IAR-HP. The pie charts represent the count of run-of-the-river (red) and storage (blue) hydropower systems within each province. The size of the pie chart is proportional to the total installed power in each province. Each province is numbered according to the ID's listed in Table 4.

facilities, totaling 21, while Aosta ranks first in the number of RoR systems, with 29 systems. Moving towards flatter areas, the amount of LHS generally decreases, although it might be constituted by a few significant run-of-the-river plants handling large streamflows.

IAR-HP also includes specific structural and operational information for 156 reservoirs (most of which serve the primary purpose of hydropower production), ensuring high accuracy and reliability in the data. These reservoirs directly or indirectly impact the operation of the hydroelectric plants. Fig. 3 summarizes the main statistics for reservoirs in IAR-HP. A total active storage volume of 2312.2 million cubic meters is composed of a large fraction of very small reservoirs (more than 80 with a volume lower than 5 $Mm^3$), realizing slightly more than 5% of the total volume: these reservoirs usually exploit high heads to achieve a good power outputs, and most of them handle relatively low flows, performing daily regulation activities. Many





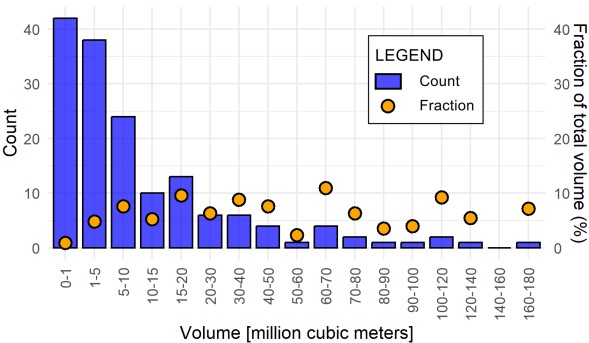

**Figure 3.** Statistics of reservoirs present in IAR-HP: blue bars represent the count of reservoirs falling within each volume class. Orange dots represent the relative contribution of each volume class, expressed in percentage, to the total volume of reservoirs.

medium-sized reservoirs perform daily-to-monthly regulation, depending on their regulation capacity (ratio between average inflow and active storage). Finally, four very large reservoirs make up for around 25% of the total active storage. Given the average inflows that usually fill alpine catchments (often not exceeding 10 $m^3/s$), most reservoirs with more than 20 $Mm^3$ active storage can perform seasonal regulation activities. In some peculiar/drought situations, reservoir water resources can also be used for irrigation purposes, conflicting with hydropower production. However, since this occurrence is very minor

within the domain of IAR-HP and often limited to medium-large reservoirs, collecting information on the agricultural water usage was deemed beyond the scope of IAR-HP.

## 2.2 Dataset validation

Our contribution aims to provide a valuable and reliable source of information for hydrological modeling endeavors. This, of course, entails validating the information provided herein. The scarcity of homogeneous and openly accessible sources

of information, however, makes the validation of the dataset in its strict sense unfeasible. Hence, an alternate approach was adopted: the validation goal is to reproduce the observed hydropower production, resorting to a hydrological modeling exercise that includes the entire spatial extent of IAR-HP. This section will highlight the setup and key characteristics of the modeling framework.

### 2.2.1 Hydrological Model

As anticipated, IAR-HP was initially designed to comply with the data requirements of HYPERstreamHS (Avesani et al., 2021), a holistic, distributed hydrological model that includes routines for explicitly simulating the alterations in streamflow related to the functioning of man-made hydraulic facilities. HYPERstreamHS thus represents the most consistent model choice for validating IAR-HP. A similar validation exercise restricted to the Adige catchment was performed in Galletti et al. (2021), showcasing the benefits of including thorough hydraulic and management information in large-scale hydrologic assessments.

The model first performs vertical water mass balance (i.e., precipitation, snow accumulation/melt dynamics, evaporation and



partitioning into infiltrated water and surface runoff) at the level of discrete spatial units named macrocells. Surface runoff is then routed to the hydrologically nearest downstream node by means of a widthfunction-based instantaneous unit hydrograph scheme (Piccolroaz et al., 2016). After reaching the first downstream node, runoff enters the network of modeled nodes and links where it undergoes different streamflow partitioning and routing routines (i.e., natural in-stream routing or flow partition-

ing to model the effect of hydropower infrastructures). HYPERstreamHS embeds a dual-layer MPI parallel computing scheme that allows it to undertake challenging computational endeavors, such as calibrating the 12 hydrological parameters of the model (Avesani et al., 2021).

### 2.2.2    Meteorological and land cover data

The hydrological model relies on spatialized input information: DEM, land cover, daily gridded precipitation and tempera-

ture.

The DEM is used to compute the hydrologic drainage network and to attribute average height to each discrete macro-cell. For this case study, we adopted the 30 m resolution EU Digital Elevation Model (EUDEM, https://www.eea.europa.eu/data-and-maps/data). Land cover information comes into play for computing potential evapotranspiration (PET) according to Hargreaves and Samani (1982). Land cover classification was extracted from the CORINE 2006 database (https:

//www.eea.europa.eu/publications/COR0-landcover).

The Alpine precipitation gridded dataset (APGD,  Isotta et al., 2014) was adopted as the precipitation input dataset: APGD is a spatial analysis of precipitation events over the European Alps. APGD is available at a 5x5km horizontal resolution dataset and contains daily precipitation from 1971 to 2008. The estimations are sourced from more than 8500 gauge stations across the region, providing dense observational data Isotta et al. (2014), making APGD one of the most accurate gridded products

available over the Alpine domain. Daily mean, minimum and maximum temperature were retrieved from the COSMO-REA6, a reanalysis dataset containing hourly temperature time series from 1995 to 2019, (Bollmeyer et al., 2015), which was shown to have superior performance compared to other datasets in the Alpine Region (Scherrer, 2020). Consequently, the modeling time window was limited to the overlapping portion of the two datasets, from 1995 to 2008, for a total of 13 years.

### 2.2.3    Gauging stations data

HYPERstreamHS hydrological parameters are often calibrated in an attempt to optimize the reproduction of observed flows. The stream gauging stations' daily streamflow time series were obtained using the free-to-use Italy Central Hydrological Information System portal (http://www.hiscentral.isprambiente.gov.it/hiscentral/default.aspx). The database was updated with collected data from individual regional environmental agencies. This endeavour resulted in a daily observational streamflow dataset covering from 1845 to 2019 and totalling 486 stream gauges throughout the Italian Alpine Region. It should, however,

be noted that data quality is, albeit generally good, not checked, and time series need to be manually inspected to ensure their reliability.





#### 2.2.4 Hydropower production data

TERNA, the Italian electrical grid manager (Terna, 2024), supplied hydropower production data upon request. Data are organized into monthly production time series and aggregated at the province level. The provided dataset belongs to a period
of 15 years starting from 2000 without any gaps. It is separated into two information categories based on the 3 MW installed power classification threshold explained earlier.

The TERNA report covers the period from 2000 to 2015, extending beyond our modeling duration of 1995 to 2008. Hence, the validation of hydropower production was only related to production in the overlapping window of 2000-2008. Following the approach detailed in (Galletti et al., 2021), hydropower production data is also used to derive reservoir operation rules for
each province. Even in this case, only the pattern derived from the 2000-2008 monthly averages was considered.

#### 2.2.5 Hydrological model calibration

Before running the hydrological simulation, the model parameters were calibrated to optimize hydrological modeling throughout the domain (model parameters are detailed in Piccolroaz et al., 2016; Avesani et al., 2021). Given the size of the domain, we reckon that a single set of parameters can not realistically represent the entire area, therefore we subdivided
the domain in several mesoscale watersheds, which is depicted in Fig. 4. The watershed boundaries were defined balancing the need for a manageable number of watersheds, (hence limiting the computational burden of the calibration), and that of supporting each of them with observed streamflow records meeting reasonable quality standards (e.g., continuous data, minimal impacts from nearby power plants, realistic low flow records). This led us to define a final set of 9 watersheds, 8 of which were calibrated relying on a single gauge station located in the downstream region of the catchment. For the rightmost area, we ran
into the issue that no reliable streamflow record is available for the region of Friuli-Venezia Giulia (the region is well-known for river braiding, see e.g. Bertoldi et al. (2010)). In this case, we resorted to a multi-site calibration of the closest catchment, Piave, located in the neighbouring region of Veneto, considering three headwater stream gauges that presented good streamflow records to improve the spatial representativeness of the parameters. The stations used for the multi-site calibration of this watershed (Piave river + Friuli-Venezia Giulia region) are located in dark-green shaded area in Fig. 4.

The result of the calibration is summarized in Table 3. The calibration performance is excellent over almost the entire domain, achieving very good performances in reproducing daily flows in six watersheds out of nine (NSE > 0.7, according to the classification introduced by Moriasi et al. (2007)), and satisfactory performances in two of them, with NSE above 0.5. The Brembo basin exhibits the worst performance with a NSE of 0.44, an occurrence that we attribute to the small size of the watershed (around 200 $km^2$), whose actual precipitation patterns are harder to accurately capture, subsequently affecting
streamflow reconstruction.

HYPERstreamHS embeds several calibration algorithms which have been refactored for parallel computing: we adopted Particle Swarm Optimization (PSO) Kennedy and Eberhart (1995) for its ability to handle both single- and multi-site calibration. The optimal parameter sets were computed by iteratively maximizing the resulting Nash-Sutcliffe Efficiency (NSE) index Nash and Sutcliffe (1970) at the selected gauge stations for each individual watershed; in the case of the multi-site calibration,





**Table 3.** Calibration efficiency in the nine watersheds considered in this study. Watershed name refers to gauged rivers mentioned in Fig. 4

| Watershed | Calibration NSE |
|-----------|-----------------|
| Dora | 0.88 |
| Toce | 0.92 |
| Ticino | 0.85 |
| Adda | 0.67 |
| Brembo | 0.44 |
| Chiese | 0.69 |
| Adige | 0.84 |
| Brenta | 0.73 |
| Piave | 0.72 |

the average of the three NSEs was maximized. The PSO approach was executed for each area using 100 runs and 100 particles, following the procedure extensively detailed in Avesani et al. (2021): this setup ensures a thorough exploration of the parameters' space. The calibration was performed from 1996 to 2008, excluding the first year as a spin-off.

### 2.2.6 Assumptions for the validation of the hydropower production

The simulated plant production was validated by comparing the simulated total annual energy production with historical
recorded values of hydropower production.

In particular, within TERNA's report, the average production of some provinces is aggregated and provided as a single record. To ensure a proper comparison with TERNA's report and validate our modeled hydropower production, we thus performed the same aggregation on the following provinces:

- Gorizia-Udine (GU): these two provinces are reported jointly in TERNA's report.

- Milano-Pavia (MP): the Pavia record includes data from the Monza province, established in 2009 and previously belonging to the Milano province. For consistent comparison, since there is no trace of this shift in TERNA's report, we merged these two provincial records and validated them as a single one.

- Alessandria-Biella-Novara (ABN): these three provinces are reported jointly in TERNA's report. No LHS are reported for the Novara province in IAR-HP but we keep this aggregation for consistency with the observations.

- Verbano-Cusio-Ossola (VCO): is actually one province named after three cities in it, therefore no actual aggregation is performed here. We will, however, shorten its name to VCO for brevity.

- Brescia-Mantova (BM): The province of Mantova is not reported in the TERNA report. therefore, we credit the production of its single plant to the neighbouring province of Brescia.





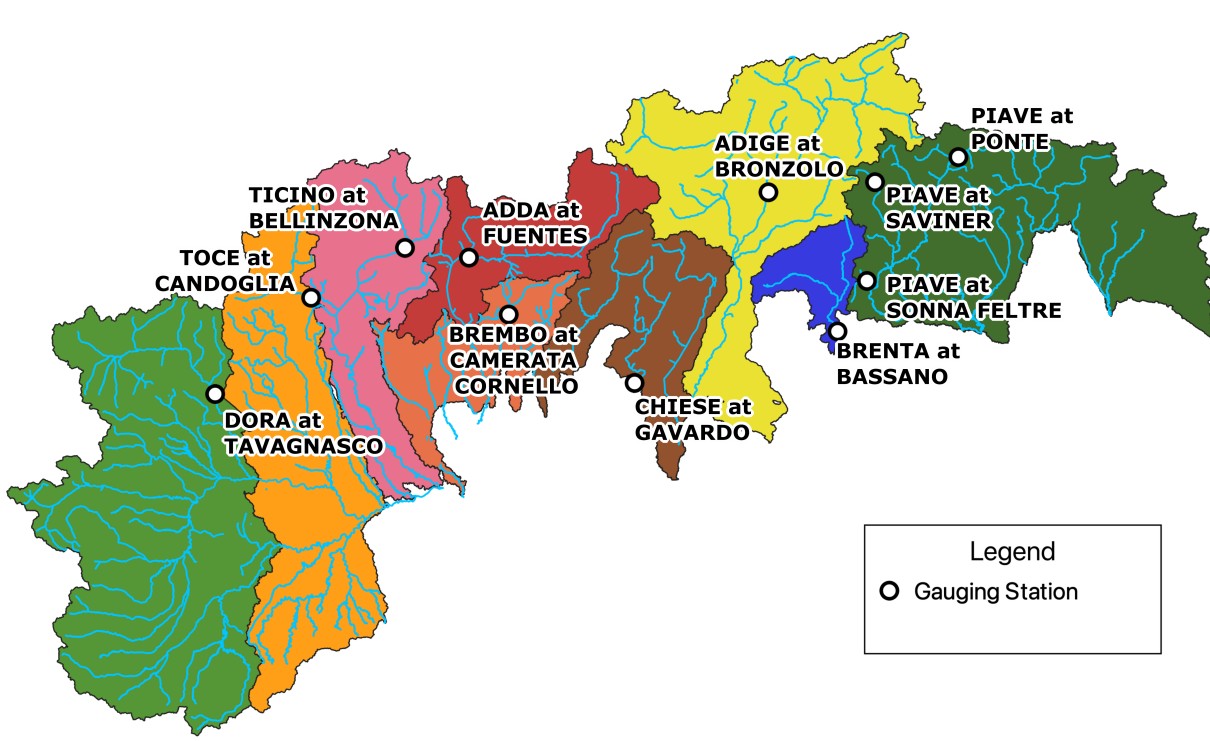

**Figure 4.** Study area watersheds with their representative gauging stations.

This aggregation procedure reduces the initial 25 provinces for which IAR-HP lists at least one LHS, to a new total of 21, for
which results will be presented. It is worth mentioning that the Trento province represented the only peculiar case of a province
crossing several watersheds, namely the Chiese, Adige, and Brenta (see Fig. 3 and 4). Therefore, the hydropower production
of plants pertaining to said watersheds was computed by adopting their respective hydrological parametrization.

Furthermore, the current version of HYPERstreamHS (Avesani et al., 2021) is not equipped to explicitly simulate the water
being pumped back upstream in pumped storage hydropower (PSH) and mixed pumped storage hydropower (M-PSH) systems:
since pumping constitutes a major component of the available volumes for PSH, the inability to model this mechanic would
affect modeled production significantly. This effect is proportionally lower in the case of M-PSH, where water is pumped
back less often. To avoid this bias and for the sake of consistency, we chose to consider the long-term average hydropower
production of both PSH and M-PSH as declared by each plant's owner (https://www.enel.com), in place of their simulated
counterparts, when validating our results. This assumption was applied to a total of 13 LHSs.



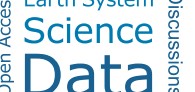

## 3   Simulation of hydropower production

The ensuing paragraphs will extensively describe the hydropower production modeling outcomes in each province, comparing them to historical data and detailing peculiar situations that might have been relevant to the final outcome. The results are summarized in Fig. 5, and additional information regarding the detailed breakdown of the final provincial grouping can also be found in Table 4.



**Figure 5.** Model performance in simulating provincial hydropower production: bar height (left axis) represents observed annual average hydropower (2000–2008) for each province; bar colors (right color scale) and numbers above bars indicate the absolute relative error computed as $|HP_{obs} - HP_{sim}|/HP_{obs} \times 100$. Abbreviations refer to the assumptions detailed in Sect. 2.2.6.

The Aosta province accounts for 35 LHS, comprising 6 storage and 29 Run-of-the-River plants. The total installed capacity in the province is approximately 890 MW, ranking it as the fifth-highest province within IAR-HP in terms of hydropower production. The two largest installed plants in the province are Avise and Valpelline, with an installed capacity of 126 MW

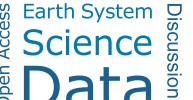

**Table 4.** Comparison of average observed and simulated hydropower production (HPP) in each province during the period 2000-2008. Plants count as of 2015/12/31 provided by Terna compared to the plants in IAR-HP. Superscript letters near abbreviations refer to the grouping assumptions detailed in the previous Section: a) Brescia-Mantova, b) Verbano-Cusio-Ossola, c) Gorizia-Udine, d) Milano-Pavia, e) Alessandria-Biella-Novara.

| No. | Province | Observed HPP [GWh/year] | Simulated HPP [GWh/year] | Relative Error [%] | IAR-HP Plant Number | TERNA Plant Number |
|---|---|---|---|---|---|---|
| 1 | SONDRIO | 4821.36 | 4290.32 | -11.01 | 38 | 44 |
| 2 | BOLZANO | 4593.49 | 3892.22 | -15.26 | 28 | 44 |
| 3 | TRENTO | 3344.59 | 3329.30 | -0.45 | 30 | 37 |
| 4 | BM$^a$ | 3202.44 | 3347.26 | +4.52 | 31 | 36 |
| 5 | AOSTA | 2713.61 | 2870.85 | +5.79 | 35 | 34 |
| 6 | CUNEO | 2455.53 | 2369.42 | -3.50 | 21 | 29 |
| 7 | VCO$^b$ | 2006.13 | 1761.06 | -12.21 | 32 | 35 |
| 8 | BELLUNO | 1755.35 | 1733.66 | -1.23 | 20 | 19 |
| 9 | TORINO | 1734.43 | 1538.36 | -11.30 | 32 | 36 |
| 10 | VARESE | 1354.54 | 1316.60 | -2.80 | 5 | 5 |
| 11 | BERGAMO | 649.25 | 468.26 | -27.87 | 20 | 26 |
| 12 | VERONA | 634.75 | 804.22 | +26.69 | 6 | 7 |
| 13 | TREVISO | 609.08 | 837.69 | +37.53 | 6 | 5 |
| 14 | PORDENONE | 593.13 | 741.19 | +24.96 | 13 | 13 |
| 15 | GU$^c$ | 578.12 | 590.67 | +2.17 | 6 | 5 |
| 16 | MP$^d$ | 534.35 | 561.79 | +5.13 | 8 | 7 |
| 17 | VICENZA | 222.89 | 142.06 | -36.26 | 5 | 4 |
| 18 | COMO | 108.05 | 129.27 | +19.63 | 2 | 3 |
| 19 | ABN$^e$ | 54.30 | 81.68 | +50.42 | 3 | 4 |
| 20 | VERCELLI | 51.20 | 18.47 | -63.91 | 1 | 4 |
| 21 | LECCO | 50.48 | 33.87 | -32.89 | 1 | 3 |

and 130 MW, respectively. From 2000 to 2008, the province has a historical production of nearly 2700 GWh/year. Notably, there are no pumped-storage plants in the province, and the simulated production of this province was about 2870 GWh/year, matching the observed value very closely.

Torino is home to 11 storage and 21 run-of-the-river plants, totaling 795 MW in installed power. The two largest plants, Venaus and Pont Ventoux, have installed powers of 240 MW and 150 MW, respectively. Pont Ventoux and Telessio are M-PSH systems; following Sect.2.2.6, their simulated production was replaced by their nominal declared production (350 GWh/year and 40 GWh/year, respectively). Historical production in Torino is approximately 1730 GWh/year, with a simulated production of about 1540 GWh/year. The missing production (about 11% with respect to observations) in this province is likely due to





an imperfect representation of the flows feeding the hydropower plants, as the Dora river is almost completely located in the Aosta province. Moreover, four stations are missing in this province compared to TERNA's report (see Table 4), which might have contributed to this deficit.

Cuneo is home to 21 LHS, consisting of 7 storage and 14 run-of-the-river facilities, with a combined installed power of
approximately 1547 MW. Notably, the province houses the largest pumped storage hydropower system in the IAR-HP (Entracque). The system consists of two separate PSH groups (named Chiotas and Rovina), with a combined installed power of nearly 1200 MW, representing 78% of the province's total capacity and totaling an average annual production of 1460 GWh/year: being a pure-pumping system, the historical production for this system was considered in place of its simulated counterpart, as previously explained in Sect.2.2.6. Cuneo's average annual production is around 2455 GWh/year, and its sim-
ulated value is very close, equaling about 2370 GWh/year.

VCO is home to 32 large hydropower systems with 11 storage and 21 run-of-the-river plants, most of which have relatively small installed power. This province's average installed power per system is approximately 22MW; the biggest plant is located in Caderese and has 70MW installed power. Overall, VCO's total installed power stands at approximately 700 MW. The long-term average production of the province is about 2006 GWh/year. Also, the simulated value for this province is about 1750
GWh/year. The 12.2% deficit in simulated production is likely due to a combination of a few missing plants (three) and a sub-optimal representation of reservoir operation, given the very good hydrological calibration results for this area (Toce basin, see Table 3).

ABN has three LHS, all of which are of the storage type, with a total installed power of 28 MW and historical production of approximately 54 GWh/year. The simulated value for this province was about 80 GWh/year.
Vercelli hosts one storage LHS with an installed power of about 4.25 MW, contributing to the province's average annual production of approximately 51 GWh. At the same time, this province's simulated value was 19 GWh/year, with a significant underestimation.

We attribute the differences of both ABN and Vercelli to an incorrect representation of streamflows; however, it should be noted the high relative errors (+50% and -64%, respectively) are associated with small differences in absolute terms of the
hydropower productions, respectively 27 Gwh/year and 32 Gwh/year for ABN and Vercelli. These mismatches are minimal if compared to the total production in IAR-HP.

Varese province is home to 2 storage and 3 run-of-the-river plants accounting for an installed power of approximately 1050 MW. Among these, the Roncovalgrande plant stands out, with an installed power of 1000 MW. As per the other PSH, the long-term average production of Roncovalgrande (1000 GWh/year) was considered in place of the simulated one. During the
period 2000-2008, the Varese province produced about 1350 GWh/year, while the simulated output is 1300 GWh/year. The results over this region were highly satisfactory, with the model output achieving a 97% match with historical values.

Following the aggregation explained in Sect.2.2.6, MP results in 6 run-of-the-river and two storage plants, with a total installed power of 95 MW. Historical production is approximately 530 GWh/year, with simulated production reaching 560 GWh/year. Despite its relatively small installed power, Milano's production is notable due to the sustained high flow rates
feeding its LHS, with an average capacity of 105 $m^3$/s.



Bergamo hosts 20 large hydropower systems, including 7 storage and 13 run-of-the-river plants, with an installed capacity of over 220 MW. Three major plants of this province, namely, Carona, Bordogna, and Dossi, with an average installed power of 46MW, play an important role in the whole average annual production in Bergamo, which is about 650 GWh/year. At the same time, the simulated yearly output is almost 470 GWh/year. It is worth mentioning that IAR-HP is missing six LHS compared
to TERNA's report, likely contributing to the 28% deficit achieved by our simulation.

Como has 1 storage and 1 run-of-the-river plant, with an installed power of 28 MW and a historical production of 100 GWh/year, while the simulated value is 129 GWh/year.

Lecco has a single storage plant with an installed capacity of 15 MW and an annual production of 50 GWh/year, while the simulated counterpart amounted to 34 GWh/year.

The poor results in these provinces are, once again, probably influenced by a combination of a reduced plant count and imperfect representation of streamflow feeding the reservoirs.

With 38 LHS plants, Sondrio has the highest number of installed hydroelectric plants in the IAR-HP. This includes 19 storage plants and 19 run-of-the-river plants, with a total installed capacity of 2150 MW. Sondrio is also home to four of the largest installed LHS in IAR-HP, namely Grosio, Premadio, Lanzada, and Mese, with installed powers of 431 MW, 245 MW,
188 MW, and 173 MW, respectively. There are also two M-PSH systems installed in this province, namely, Campo Moro and Zappello, for which the nominal declared production was considered (29.88 GWh/year and 18.36 GWh/year, respectively). This province's average production is about 4800 GWh/year, making it the highest producer among all the provinces. The simulated production value of this province is around 4290 GWh/year. We notice that IAR-HP is missing six LHS compared to TERNA's report; furthermore, this might be compounded by precipitation undercatch issues in the highest reaches of this
watershed, likely influencing the final output of our simulation.

BM houses 31 LHS consisting of 8 storage and 23 run-of-the-river systems. Most of the installed power comes from three PHS with an installed power of 1680 MW, notably the Edolo hydroelectric plant with an installed power of 977 MW. The total installed power in the region is around 2060 MW. The average annual production is 3200 GWh/year, whereas the simulated production is about 3340 GWh/year, a very satisfactory outcome given the productivity of this province. For three M-PSH, namely
Edolo, Gargnano, and San Fiorano, the long-term average yearly production values of 1075 GWh/year, 198.5 GWh/year, and 559.7 GWh/year, respectively, were considered for validation instead of their simulated counterparts, as described in Sect.2.2.6.

The province of Bolzano has 28 LHS, comprising 13 storage and 15 run-of-the-river plants. The total installed capacity is nearly 1400 MW, with individual plant capacities ranging from 6.4 to 230 MW. Following Sect.2.2.6, the simulated value for the PSH plant of Pracomune was considered equal to its declared nominal production (15 GWh/year). The province's average
annual historical production is about 4600 GWh, while the simulated production rate shows a satisfactory similarity of 3900 GWh/year. We notice that IAR-HP is missing sixteen LHS compared to TERNA's report, the largest deviation from TERNA's report for a single province (see Table 4), which is accompanied by a relevant impact on the accuracy of the simulations (-15.2% from the observed value).

Trento has 30 LHSs, with 9 run-of-the-river plants and 21 storage types, totaling 1610 MW installed power, of which
1500 MW are contributed by storage type hydropower. Plants in Trento have capacities ranging from 5 to 350 MW. Santa





Massenza-Molveno is a pure PHS, while Riva del Garda 1 is a M-PSH; for both of them the long-term average production of 600 GWh/year and 124.8 GWH/year, respectively, were considered. This province's long-term historical production is approximately 3340 GWh/year, and its simulated production is about 3330 GWh/year, with a remarkable similarity of 99% compared to historical production.

Verona province does not present storage hydropower systems; all 6 LHS in the dataset are run-of-the-river systems exploiting the lowland high flows of the Adige river. The diversion channel systems installed for this province have large capacities, averaging 130 $m^3/s$. This high average flow rate allows the province to have a long-term average production of 635 GWh/year, while the simulated results indicate a production rate of 800 GWh/year. In this case, the overestimation is likely due to agricultural withdrawals not being modeled, which are non-negligible in this area, leaving more water available for production.

Vicenza has 3 run-of-the-river plants and a single storage plant, with a total installed power of 44 MW. The historical production rate for this province is reported at 220 GWh/year, with the storage plant contributing nearly half of the total power generation. The simulated production value for Vicenza is 140 GWh/year, a deficit that is chiefly attributable to one missing plant, possibly compounding with a lacking representation of available flows.

In the province of Belluno there are 11 storage and 9 run-of-the-river plants, fed by the upper portion of the Piave river
catchment. The average installed power in this province is around 17 MW, with the exception of the Soverzene plant, which has an installed power of 210 MW. The total installed power in Belluno is 540 MW. The long-term average annual production is 1755 GWh, while the simulated production rate for this province is 1730 GWh/year, benefiting from both an accurate representation of LHS in the area and of the available flows, improved by the multi-site calibration framework, as highlighted in Galletti et al. (2021).

In Treviso province, there is one storage and 5 run-of-the-river plants, with a total installed power of 360 MW. Fadalto Nuova and Nove, with a combined installed power of 320 MW, are the leading plant producers in the province. Since Fadalto Nuova is a M-PSH, its production was accounted equal to the declared nominal production of the system (344.8 GWh/year). The capacity of the diversion channels for RoR plants in Treviso ranges from 14 to 135 $m^3/s$, allowing for a long-term average production of 610 GWh/year. The simulated results overestimate the production, yielding about 837 GWh/year. We attribute
the poor performance to a weak representation of the flows and notice that IAR-HP contains one more LHS than those reported from TERNA for this province.

There are 6 power plants in GU, comprising 4 run-of-the-river plants and 2 storage plants. Approximately 70% of the installed power in this area comes from the Somplago plant, with an installed power capacity of about 173 MW. These provinces' combined installed power capacity is 250 MW, leading to an annual production of 580 GWh, with a simulated value of 590
GWh.

Pordenone features 13 LHS, 4 storage, 9 run-of-the-river plants, and 360 MW installed power. The intake capacity of the RoR plants averages 25 $m^3/s$ ranging from 7 to 30 $m^3/s$. Pordenone has an average historical annual production of 590 GWh/year production, and the simulated value of this province is about 740 GWh/year, likely because this area is not hydrologically similar to the Piave catchment, from which it borrowed its hydrological parametrization.



Overall, our results show high accuracy in reproducing observed hydropower production. Of 32.1 TWh/year, 30.9 TWh/year were correctly reproduced by adopting IAR-HP in the HYPERstreamHS framework, with a ratio of 96.2%. Results were very satisfactory in all highest-producing provinces, for all of which the relative error was within 15%. The influence of compensating errors (overestimation vs. underestimation) was limited, with an average of relative RMSE of 14.8% across all provinces.

## 4    Discussion

The level of accuracy achieved in the validation of IAR-HP indicates that the dataset effectively captures the spatial distribution and operational characteristics of LHS and is well suited for use in hydrological modeling endeavors in the study region. Such precision facilitates reliable modeling outcomes, thereby supporting water resource planners and policymakers in decision-making processes related to energy production and distribution in the Italian Alpine region. Despite these positive aspects,

several challenges have been identified. We notice how (see Table 4) our simulated production is usually negatively biased, especially in the provinces with the largest production and plant count, often characterized by high-head LHS located in upstream catchments. We attribute this to a combination of i) IAR-HP not reporting some smaller LHS as opposed to those reported officially by Terna, ii) hydrological calibration and undercatch affecting the amount of water available in the upper portions of these catchments, often contributing to the largest share of hydropower production, and iii) simplified reservoir

operation schemes, leading to sub-optimal production performance. Conversely, positive bias (less relevant in absolute terms) emerged mainly in provinces with abundant low-head, high-flow LHS. in this case, we attribute the bias to i) slight head differences between our reported head (computed according to the DEM) and the real one combined with ii) bias in the large flows handled by these plants.

We group the aforementioned issues under two categories, namely *reporting issues* and *hydropower modeling challenges*,

on which we will focus on in the ensuing sections. Finally, we would like to compare IAR-HP with two recent contributions.

### 4.1    Reporting issues

The first issue concerns reporting issues resulting in the small mismatch between the number of facilities reported in the TERNA database and those collected in IAR-HP (see Table 4). We identified a few main causes for this. A notable issue is the attribution of hydropower plants located near administrative borders to the correct province. Indeed, ambiguities often arise

regarding the jurisdiction these plants belong to, which in turn opens up to potential misallocation of production data across provinces. In addition, reporting discrepancies are also present: official reports may list multiple production groups within a single facility separately (e.g., due to different units being built at different times), whereas our dataset considers these groups as a single entity. On the one hand, we can't reconstruct exactly which productive units are counted as multiple, but on the other hand, this counting convention is only relevant in terms of count. At the same time, the installed power for each plant

is reported correctly and confirmed by our validation exercise. Finally, we acknowledge that some plants, likely smaller ones close to the LHS power threshold of 3MW, might be missing from our dataset, although the results presented in the validation





exercise make us confident that the major ones are correctly reported. This is somewhat confirmed by the fact that the plant count gap between TERNA and IAR-HP is largest in the highest-producing provinces (where it is common to have several productive units in the same building, usually originating by incremental development of the facilities).

## 4.2 Hydropower modeling challenges

The second challenge concerns the uncertainty associated to hydropower production modeling. Management practices, particularly those concerning storage and pumped storage hydropower systems, introduce significant uncertainties that can lead to differences in the exact timing of water usage, in turn affecting the hydraulic head and cascading onto energy production. Furthermore, the structural geometry of these systems, including the precise location of generators, can impact the head and, consequently, power generation efficiency. For instance, generators located below the surface might result in an increased effective head compared to that obtained through DEM differences, altering expected outputs. These structural uncertainties affected the output of our modeling activity to some extent, although once again, we are reassured by the quality of our results. It is also worth noticing that agricultural water withdrawals are not modeled in HYPERstreamHS, effectively granting more water during the irrigation season, especially in lowland catchments where this kind of water use is more prominent. Finally, the accuracy of hydropower modeling is strongly related to the input data, chiefly precipitation but also temperature and evapotranspiration. These variables all exert macroscopic effects on the water balance, defining the volume of water available to hydropower systems and thus affecting hydropower production. Moreover, hydrological parametrization is critical in determining the timing and quantity of water available for hydropower generation. Variability in these parameters can lead to deviations in model predictions, underscoring the importance of precise data collection and parameter estimation in enhancing model reliability. We do however believe that the very good hydrological calibration results ensured a realistic water balance throughout our calibration exercise.

## 4.3 Similar contributions

In this section, we put IAR-HP in relationship with two very valuable contributions that were recently published, from Evangelista et al. (2024, preprint) and Catania et al. (2024). Evangelista et al. (2024) gathers information about the attributes of 528 large dams in Italy, including dam characteristics, geographic coordinates, structural features, and upstream catchment data. The study also integrates climatological data, land cover, and NDVI values, allowing for a comprehensive hydrological and environmental assessment of each dam's catchment area. Evangelista et al. (2024) reports 184 hydropower dams in the IAR-HP domain, while 156 are recorded recorded in IAR-HP. This difference arises from two factors: firstly, Evangelista et al. (2024) reports the presence of multiple dams where they are present, although no information on their stage-storage relationship is provided. Conversely, IAR-HP only adopts the main dam, for which MIT provided a stage-storage relationship. Indeed, The availability of a stage-storage curve at each dammed location is an essential prerequisite for correctly modeling dam storage dynamics in HYPERstreamHS, hence our recording choice.

Catania et al. (2024) compiled a database of the programmable (Reservoir- and pumped-storage) LHS in the Italian Alps. They recorded information on power capacity (for pumping storage plants, both charging and discharging), energy capacity,



head, volume of the basin, geographical coordinates. In the area of IAR-HP, Catania et al. (2024) reports 128 programmable LHS, as opposed to the 129 reported in IAR-HP; furthermore, upon deeper examination, some of the plants reported in Catania et al. (2024) seem not to exceed the 3MW threshold set by Italian legislation for LHS. Finally, neither of the two aforementioned contributions includes (due to their scope) information about:

  – run-of-the-river LHS in general,

– diversion channel location and abstraction capacity,

  – environmental flow requirements for both reservoirs and diversion channels, and

  – reservoir operating rules.

This information is, in our opinion, crucial for performing thorough assessments of the water-energy nexus and is all included in IAR-HP (see as an example the inset in Fig. 1). Furthermore, IAR-HP information was explicitly validated through a
hydropower production modeling exercise. We stress that a direct comparison among these three datasets is not possible, nor would it appropriately credit the value each of them brings. Rather, we see IAR-HP as a very valid addition to a widely acknowledged gap in the (Italian) water-energy modeling community, able to effectively complement information from other sources. IAR-HP holds significant potential for supporting broad water-energy nexus studies, making it a valuable asset for integrated resource planning and sustainable development efforts. It can serve as a pivotal component in scenario analyses
that range from simulating the impacts of climate change on hydropower production, to evaluating the effects of varying water management strategies and/or environmental policies, and to assessing the resilience of energy systems under different hydro-meteorologic conditions.

## 5  Conclusion

We believe that IAR-HP represents a very valuable contribution to the water-energy modeling community, especially that
working on the Italian Alpine Region: the pressing need for this kind of information is confirmed by similar openly available dataset being completed and published recently. Nevertheless, fully understanding the limitations, as well as the assumptions that were made in compiling IAR-HP, can help to maximize the outcomes of its adoption in further research, as well as directly improving the dataset.

   While IAR-HP has demonstrated robust performance in hydropower modeling, coherently reconstructing 6.2% of observed
hydropower across 21 provinces in the Italian Alps, we underscore the necessity for continual refinement and validation of its contents. Addressing geographic, reporting, and structural uncertainties will help increasing the quality of the dataset per se, while enhancing the accuracy of input data and parametrization is equally essential for achieving reliable modeling outcomes and ensuring their applicability across diverse regions. The dataset's potential for facilitating comprehensive water-energy nexus assessments further highlights its value as a tool for advancing integrated resource management and planning.



*Data availability.*

The IAR-HP dataset is shared freely on Zenodo, and data are available for download at https://doi.org/10.5281/zenodo. 14040999, (Galletti et al., 2024). Data include LHS characteristics, as well as (simulated) Minimum Ecological Flow requirements and turbine discharge scheme adopted for Reservoir hydropower plants. Data are stored in the form of Excel (.xlsx) tables and separate shapefiles (.shp) for each node type (Plant, Intake, Reservoir) are provided; R and Python scripts that were used to preprocess data into inputs for HYPERstreamHS, as well as the model itself, are available upon request to the authors. The dataset is released under the Creative Commons Attribution 4.0 International license.

*Author contributions.*

A.G.: Writing - original draft, Conceptualization, Methodology, Data Collection, Software development, Writing - Review and Editing, S.Z.D.: Writing - original draft, Data curation, Formal validation analysis, Visualization, Writing - Review and Editing, B.M.: Conceptualization, Writing - Review and Editing, Funding acquisition, Supervision, Resources.

*Competing interests.*

The authors declare that they have no conflict of interest.

*Acknowledgements.* This research received financial support by the Energy oriented Centre of Excellence (EoCoE-III), GA number 101144014, funded within the EuroHPC JU framework of the European Union. This research also acknowledges the Italian Ministry of Education, Universities and Research (MUR), in the framework of the project DICAM-EXC (Departments of Excellence 2023-2027, grant L232/2016. Andrea Galletti was partially supported by the RETURN Extended Partnership which received funding from the European Union Next-GenerationEU (National Recovery and Resilience Plan – NRRP, Mission 4, Component 2, Investment 1.3 – D.D. 1243 2/8/2022, PE0000005). This paper was produced while Soroush Zarghami Dastjerdi is attending the PhD programme in PhD in Sustainable Development And Climate Change at the University School for Advanced Studies IUSS Pavia, Cycle XXXVIII, with the support of a scholarship financed by the Ministerial Decree no. 351 of 9th April 2022, based on the NRRP - funded by the European Union - NextGenerationEU - Mission 4 "Education and Research". Bruno Majone also acknowledges support from "iNEST (Interconnected Nord-Est Innovation Ecosystem)" project funded by the European Union under NextGenerationEU (PNRR, Mission 4.2, Investment 1.5, Project ID: ECS 00000043).



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
