# Peer review of "Comprehensive inventory of large hydropower systems in the Italian Alpine Region"

_Earth System Science Data, 2024_

## Author Comment (AC1)

**Reply to Reviewers**

We thank the reviewers for taking the time to thoroughly review our manuscript and dataset and for providing constructive criticism. We believe we fully addressed each point as reported in this rebuttal document. Our interventions were in three directions: i) improving the manuscript, ii) improving the database files (Excel and shapefiles), and iii) improving the documentation associated with the database.

In this document, our replies are in blue, while the original reviewer comments are in black. Within the answers, modifications applied to the main text are further highlighted by the *italics* font, reporting them in full whenever appropriate, while only citing the corrections if they were minor. We updated the database to its version 4 (v4 on Zenodo, https://doi.org/10.5281/zenodo.14040971), including i) the improved documentation, ii) an improved version of the Excel spreadsheet, and iii) additional shapefiles delineating catchment boundaries. We also developed a new Appendix following Reviewer 1 comment 6.

At the end of this rebuttal document, we provide the new Appendix and the revised documentation of the dataset.

Reviewers Comments:
RC1:

1) The Q_avg, in Excel, was calculated from P_med. From the equation it seems to me that P_med is an energy, not a power, and the term "P" may confuse. I suggest to use P for power and E for energy.

Ans: P_med is indeed an energy (average hydropower production). We revised the nomenclature as suggested by the reviewer (comments #1, #3, #7), both in the dataset and in its occurrences in the paper (*Tables 1 and 2, and their related description in the main text*), using the symbol E for energy and W for power to avoid any ambiguity. We also took the occasion (following comment #4) to provide a more detailed documentation, where a nomenclature list is now included (nomenclature list now present on the dataset spreadsheet and on the documentation) and a more detailed explanation of all variables is provided. Moreover, to enhance readability, we added a list of abbreviations at the beginning of the paper (following Reviewer 2 minor comment #1).

2) Also, I would like to ask you about the capacity factor. I saw that in the equation the author use a coefficient of 0.8, which I believe is the efficiency, and then consider all the annual hours. If the capacity factor were considered (which could generally be around 0.35), the average flow rate during the turbine hours would be approximately 3 times higher. Therefore that average flow rate is spread throughout the year, even during periods when the system is not working.

Ans: Thank you for pointing this out, we agree with the comment. Due to the way the model is structured, the hydropower plant operates whenever water is available; this design choice is

made to compensate for our lack of knowledge of exact hydropower production schemes, which are kept confidential by plant operators (as detailed in the manuscript, as well as in https://doi.org/10.1002/wat2.1083, https://doi.org/10.1016/j.jhydrol.2021.127125 and several similar contributions). Therefore, we needed to use a datum that could be evenly distributed throughout the year, just as mentioned by the reviewer. This means that, while the model may not be very accurate on a daily or sub-daily scale, it should adequately capture the volumes on a weekly or monthly basis. We did clarify this aspect in the main text (see below), and provided additional information in an appendix which was specifically developed to address comment #6 of the same reviewer.

"[...] $Q_{AVG}$, calculated based on the long-term mean production values declared for each corresponding hydropower system, $E_{AVG}$. This approximation of the reservoir operation scheme assumes that hydropower production occurs during all available hours of the year (i.e., no capacity factor is involved in the definition of $Q_{RULE}(t)$, as this would require more specific knowledge about operation patterns).",

3) I don't understand, in Excel, what h1, h2, v1, v2 are. If they are water level and volume, the numeration should continue also for the other numbers (h3, v3,...hn, vn..)

Ans: Thank you for the feedback, and apologies for any confusion caused. Indeed $h_i$-$v_i$ represent the points of the stage-storage relationship for each reservoir, and h1 and v1 columns represent its initial (lowest) point. We originally stopped at h2-v2 because all reservoirs have different discretization count (ndisc column) of their storage, but this proved to be confusing. The dataset is now updated to a clearer format, with each column now properly labeled ($h_i$,$v_i$) to prevent confusion. Additionally, a nomenclature has been included to facilitate the interpretation of the dataset.

4) In case I wanted to reconstruct the Reservoir-Intake-Plant system from Excel: in the PLT sheet I take for example the second plant which has ID 13. Its ID_UP are 4 and 12, so I assume that they are the intakes ID4 and ID12? Then, for example, I look at intake ID 4, which in fact has ID_DOWN 13 (that very plant) and ID_UP 1 and ID_UP 3, where ID 1 is actually the reservoir, but ID3 is not in any reservoir, so is it another intake? I suggest to add a "practical example" on how to use the table in order to help the reader.

Ans: Being able to reconstruct the topology of any hydropower system in IAR-HP is a crucial aspect of our work, and we appreciate this opportunity to enhance our guidance for dataset users on how to do so. To address this, we will include a practical, illustrated example in the updated dataset documentation (see Ans. #1). We will also add a specification in the main text, in the explanation of Table 1 (geolocation and topological characteristics), see below:

"A detailed exemplification of how to reconstruct system topology is provided in the dataset documentation Section 3, (dataset available at https://doi.org/10.5281/zenodo.14040971)."

We attach the full documentation (including this exemplification) at the end of this document.

5) in the PLT sheet there are data that do not appear because they refer to other excel sheets that the authors have, for example the heads

Ans: Thanks for pointing this out. We updated a newer version of the dataset that fixes the mentioned and some other similar reference errors.

6) I suggest to try to plot Q_max (intake), Q_avg and the Q_des design flow rate of the plant calculated as Qdes=installed_power/(gamma*H*eff). What I expect is Qavg<Q_des<Qmax, where Qavg around 1/3*Qdes and Qmax around X*Qdes (X>1). This would add useful information. Then the ratio Qavg/Qdes and Qmax/Qdes may be related with the head or with the region.

Ans: We explored this interesting suggestion with reference to reservoir hydropower systems. The main result is summarized in Figure 1 of a newly developed Appendix (attached at the bottom of this document) and discussed therein. As suggested by the reviewer, Q_avg is on average about 33% of Q_des. However, no significant difference exists between Q_des and Q_max (or Q_work, in the case of reservoirs). This happens because Q_max is not the hydraulic capacity, but actually the maximum authorized flow rate (naming it capacity was an inaccuracy of ours, which is now fixed in all occurrences within the text). Hence, it is reasonable that Q_des is rather similar to Q_max. We then investigated, as suggested, the relationship between Q_avg/Q_max ratio and other characteristics of the system, such as head, active volume and regulation capacity. Albeit interesting, we deem these results to be too technical, and hence we would opt to not include them in the main text. However, as they provide further insight into the physical meaning of our recorded values, we developed a separate Appendix where these aspects are extensively presented and discussed. This can be of use to more experienced users who want to experiment with the data in IAR-HP. The newly developed Appendix is attached in full at the end of this document. Nevertheless, we plan to include the text below in the main text to further highlight the relevance of the information that will be included in the new Appendix:

"*In the Appendix, we provide further clarification on the physical and operational meaning of parameters such as Q_AVG and Q_work. We also show how their ratio, ranging from 10% to 90%, provides a rough indication of the typical operation regime of reservoir hydropower systems, with large reservoirs being operated in peaking mode (i.e., they accumulate water during off-peak periods and then discharge at full capacity and high hydraulic head to meet peak electricity demand, leveraging their large storage flexibility), and small reservoirs that continuously operate close to their maximum capacity. Furthermore, a clustering analysis conducted on the main operational characteristics of reservoirs, such as head, active volume, and regulation capacity, shows how the reservoir's regulation capacity (i.e., time to fill the reservoir assuming its average inflow) is inversely proportional to the aforementioned Q_AVG/Q_work ratio.*"

7) I suggest to add a nomenclature list, with units, in the excel file.

Ans: We added a complete list of the nomenclature in the excel file, with units for every quantitative attribute, and an explanation for each topological attribute. A more detailed explanation is also provided in the updated documentation.

8) A comparison with the existing literature in a Discussion section would be useful, for example where this database could be used to replicate some literature studies with more accurate data, or where similar analyses have been carried out.

Ans: We thank the Reviewer for raising this point. We think that your suggestion perfectly complements our existing discussion subsection (Similar contributions). We thus added a new subsection to include the suggested points, envisioning potential applications for hydrologic, energy, and eco-environmental assessments (as also suggested by the second Reviewer in the comment #2). The added subsection is provided below:

*Potential applications of IAR-HP*
*Hydrologically-based hydropower assessments in Italy have historically been limited to specific regions due to difficulties in retrieving consistent and sufficiently wide data sources (e.g., [https://doi.org/10.1002/hyp.13473, https://doi.org/10.1016/j.scitotenv.2015.05.009, https://doi.org/10.3390/cli4020016, https://doi.org/10.1016/j.envsci.2013.12.001]). IAR‑HP (together with the valuable contributions mentioned in the previous paragraph) provides an open, geographically wide and spatially consistent source of information. Its strong hydrological focus, coupled with a detailed description of LHSs, provides an improved information basis for assessing hydropower potential at both the national and continental scales (e.g., [https://doi.org/10.1016/j.scitotenv.2023.163934, https://doi.org/10.1007/s11269-022-03084-6, https://doi.org/10.1016/j.enconman.2021.114655, https://doi.org/10.1016/j.apenergy.2018.09.063]). For instance, detailed storage discretization and hydraulic head information allow for site-by-site assessments of reservoir hydropower potential, while the precise geo-location of the infrastructures may enhance the estimation of available head in run‑of‑the-river hydropower systems. Furthermore, integrating this information into a hydrological modeling framework enables a more reliable estimation of both inflows and minimum environmental flow requirements, which play a crucial role when assessing adaptation policies (https://doi.org/10.1016/j.scitotenv.2023.163934). Additionally, the high spatial resolution of IAR‑HP can support environmental studies by facilitating the correlation of hydropower-induced streamflow alterations with regional-to-local ecological assessments of some known adverse effects, such as hydropeaking (https://doi.org/10.1002/rra.4086), altered sediment transport (https://doi.org/10.1126/science.abn7980), and river network fragmentation (https://doi.org/10.1016/j.scitotenv.2023.161940, https://doi.org/10.1111/wej.12101), aiding the development of adequate mitigation strategies.*

RC2:

My major comments are:

1) The abstract could contain more quantitative features: the number of power plants in the data, total installed capacity (e.g., on L. 185), total energy production, and the share of national electricity production.

Ans: We agree with the reviewer that including quantitative insights could make the abstract more informative. We revised it to include this kind of information, adding the following sentence:

*The dataset includes detailed information about 338 LHS, with a collective installed power of 14,3 GW and an average production of 32.1 TWh/y, these LHS contribute 11.8% of the electricity generated in Italy, corresponding to roughly 80% of the national hydropower generation.*

2) 38 ff. refer to different nexus but never mention streamflow modification due to the hydropower infrastructure or negative impacts on ecosystems, such as hydropeaking, termopeaking, sediment transport, fish migration, or river network disconnectivity. In addition to the positive aspects of hydropower, these impacts on ecosystems should be mentioned in the introduction to provide a complete overview.

Ans: We are aware of the adverse environmental implications of hydropower that the reviewer mentioned, though we did not initially mention them as the focus of this section was to clarify the urge for detailed data to serve integrated modelling endeavors. We do, however, think that assessments on aspects highlighted by the Reviewer would also benefit from more detailed information on hydropower systems and their operation. We agree that those are relevant aspects and thus we revised the introduction to include such considerations. In addition, we further commented on this aspect in a newly added discussion subsection that covers potential applications of the dataset. We do, however, think that some ecological assessments might need even more detailed data than what we are providing with IAR-HP, which can only serve as a consistent starting point.

In the introduction, we added: *"On the other hand, hydropower development is known to have adverse environmental impacts - such as hydropeaking (https://doi.org/10.1002/rra.4086), altered sediment transport (https://doi.org/10.1126/science.abn7980), disrupted fish migration (https://doi.org/10.1111/wej.12101), and river network disconnectivity (https://doi.org/10.1016/j.scitotenv.2023.161940) - which integrated models should adequately capture."*
In the discussion, we added a new related paragraph, in a new subsection together with some more application outlooks following another Reviewer's suggestion (see reply to Reviewer #1 comment #8).

3) Could the dataset also be provided as a shapefile? Having the hydrological catchments that contribute to each hydropower plant as a shapefile could be relevant for future work.

Ans: We are unsure whether the Reviewer has opened the latest version of the database (v3, to the date of the review, which already included the shapefile with LHSs' coordinates). Due to a mistake on our end, the link provided in the manuscript led to v2. In v3, however, catchment shapefiles were not included. The reason for this is that the specific use of catchments is rather model-dependent (HYPERstreamHS has, for instance, embedded catchment delineation algorithms). We would also like to point out that https://essd.copernicus.org/preprints/essd-2024-387/ conducted a thorough work delineating catchments for more than 500 Italian reservoirs, which might be more akin to the resource the Reviewer is looking for (we already highlighted this valuable contribution in the Discussion). Nevertheless, in the updated version of the database, we will provide shapefiles of the catchment area contributing to each plant's reference node (i.e., the node receiving the majority of the water feeding the hydropower plant). The procedure for extracting these catchments is based on the QGIS command r.water.outlet. The necessary inputs and the pyGIS code snippet used to automate the delineation are detailed in Section 3 of the updated documentation.

4) Having information on minimum ecological flow in your data is a great benefit, especially compared to similar databases in other countries. However, I could not find the methodology for estimating these minimum flows. Maybe you can refer to this Italian paper describing the environmental flows in the different regions (Moccia et al. 2020, https://doi.org/10.4081/aiol.2020.8781 ) and describe your approach,

Ans: We thank the reviewer for the really nice reference suggestion. We were aware of the approaches reviewed in the suggested reference, and we investigated their application when developing a MEF approach for our dataset. The result of our analysis was that many of the parameters involved in the calculation were not really defined for the vast majority of Northern Italy, and were left equal to the default value of 1. Hence, almost all formulations resulted in a monthly (at best) scaling of the mean annual flow, with a scaling coefficient between 5% and 10%. For the sake of consistency, we thus defined the monthly MEF requirement at each section as 10% of the long-term average flow under natural conditions, i.e., as not affected by hydropower diversions. We reconstructed these quantities by running HYPERstreamHS with no Human Systems component active. We did, however, not include the evaluation of all regional implementations as it would require a dedicated assessment of its own, while not leading to concluding quantitative constraints.

We actually described the methodology that we adopted for computing the MEF in our dataset, though perhaps it wasn't stated clearly enough (lines 175-180 of the original manuscript). We thus revised this section to support it with the suggested reference and to make the definition and computation of MEF more clear:

*For both reservoir- and intake-type nodes, a monthly Minimum Ecological Flow requirement $Q_{(MEF)}(m)$ is defined. According to the current legislation requirements (https://pianoacque.adbpo.it/deflusso-ecologico/, in Italian), MEF should be constructed as a*

*combination of hydrological characteristics, eco‐environmental state, and the level of exploitation at each site. As detailed in Moccia et al. ([https://doi.org/10.4081/aiol.2020.8781](https://doi.org/10.4081/aiol.2020.8781)), regional administrations are currently implementing several formulations based on this general concept; however, they have not developed a thorough classification of their territories with respect to each required parameter, which makes it impossible to compute the MEF consistently. Consequently, even regional administrations often resort to a simplified estimation of MEF, fixing it at 5%–10% of the long-term average flow of each month. For consistency, we defined $Q_{(MEF,i)}(m)$ as:*

$$Q_{(MEF,i)}(m) = 0.1 \times Q_{(avg,i)}(m)$$

*with i representing each specific water withdrawal location (water intake or reservoir), m representing each month, and $Q_{(avg,i)}(m)$ being the long-term monthly average flow at each water withdrawal location. $Q_{(avg,i)}(m)$ is obtained by executing HYPERstreamHS under natural conditions (i.e., without modeling hydropower water uses).*

5) Figures 2 and 3 together seem to call for a third figure showing the share (%) of your collected hydropower plants (y-axis) contributing to the share (%) of total annual hydropower production of these plants (x-axis) to show, e.g. 20 power plants produce 50% of the total hydropower production.

Thanks for the interesting suggestion.

Considering production could be indeed challenging for two reasons: firstly, as of now long-term average production is only recorded for reservoir hydropower plants in IAR-HP. Secondly, the reference time window for this value is never standardised, as this information is retrieved from online leaflets that are only sometimes consistent (i.e., the reference window may easily vary between 5 and 30 years). Instead, we can create a similar plot referring to the share of total installed power (%). Since the meaning of this figure is very similar to what we did in Figure 2 for the reservoirs, we opted for including this information in the updated version of Figure 2, which now looks like this:

[Figure]

Figure 3: ECDF of reservoirs (square markers) and hydropower plants (round markers) against their contribution to the total active volume and installed power, respectively. The insets show the location of the reservoirs (upper) and hydropower plants (lower); a color scale based on the reservoir active volume and plant installed power is provided to facilitate locating the most relevant structures.

We also integrated the related in-text comment on the Figure with a sentence commenting on hydropower plants:

*The installed power of the 338 LHSs present in IAR-HP amounts to 14.3 GW: of these, 5 major systems constitute 25% of the total: four of them are either pumping- or mixed-pumping hydropower systems, exploiting large heads (200~900m) and handling substantial flows, and one is a reservoir hydropower system. Conversely, a large portion of smaller systems (80% of the total number of LHSs) contributes to little more than 20% of the total installed capacity.*

6) Figure 5: I'm not sure the relative error makes sense, as the smaller the annual production gets, the bigger the error becomes, as described in Table 4. I would replace it with the figure proposed in comment 5.

Ans: We respectfully disagree with the reviewer on this. Figure 5 pertains to the validation of the dataset performed by means of a hydrological/hydropower modelling exercise, while the suggested figure (comment #5) represents a nice addition to the previous description of the dataset's statistics. Our validation covers hydropower production, aggregating it at the province level, which is the finest level for which we have reliable and consistent information on hydropower production, and comments both absolute and relative modeling errors. Albeit seemingly obvious, we deemed it important to reinforce that the error committed was close to negligible in the most relevant provinces. Nevertheless, we wanted to warn the readers/users that bias in our collected information might become more relevant as plant count decreases (hence the focus on the relative bias). On the one hand, this shows that the general large hydropower systems' population and operation are captured very well (and hence, the altered streamflow volumes for e.g. ecological implications), on the other hand, it stresses the need for careful model calibration and for checking of the information that we provide, as we disclaimed in our manuscript.

7) Section 2.2 (including subsections) provides an overview of the different sources. I suggest fewer subsections, merging the information and the data source and then describing how these datasets are used for the simulations.

Ans: We merged sections 2.2.2 to 2.2.4 into a single section, now called Data sources. At the end of the subsection, we added a specification on the usage of each individual data source in our modelling endeavor, see text below:

*"Meteorological, land cover and streamflow data are used to set up, calibrate, run and validate the hydrological model, ensuring that the modeled streamflows are accurate at the watershed scale. Hydropower production data are employed in the ensuing Section 3 to thoroughly validate the outcomes of the hydropower production simulation."*

8) Section 3 provides information for each province. I suggest providing a synthesis paragraph for the entire Italian Alpine Region.

Ans: We have a similar paragraph at the end of the section in the submitted manuscript. The Reviewer is perhaps implying that it would be more helpful to have such a paragraph at the beginning of the section, instead. We agree with this view, and we moved the synthesis paragraph at the beginning of the Section, slightly modifying it as follows:

*We validated the contents of IAR-HP by modelling hydropower production over the IAR domain, as detailed in Section 2.2.6. At the aggregate level, our results show high accuracy in reproducing observed hydropower production. The inclusion of IAR-HP data in HYPERstreamHS allowed the reconstruction of 96.2% of the average annual production, 30.9 TWh/year, against a recorded value of 32.1 TWh/year. Results were very satisfactory in all highest-producing provinces, for all of which the relative error in terms of average annual production was within 15%. The influence of compensating errors (overestimation vs. underestimation) was limited, with an average relative RMSE of 14.8% across all provinces.*

And some minor comments:

1) The manuscript and the dataset are full of abbreviations, which makes it challenging to read. I propose a list of abbreviations at the beginning of the manuscript and on an added sheet in the Excel file.

Ans: Agreed. This was addressed in the revised manuscript and dataset spreadsheet. At the beginning of the manuscript a list of the main abbreviations used was added, while a complete description of the nomenclature was added in the updated documentation as well as in the initial page of the Excel file.

2) 22: I guess from the late 19th century. The late 1800s would be early for electricity production.

Ans: The sentence refers to the building of dams, the first hydropower plant built in Italy dates back to 1895 (Giorgio Bertini hydropower plant, which started commercial activities in 1898). However, we revised the sentence to "*from early 1900s*" which surely fits a larger share of Alpine hydropower development.

3) 76: The title should be "Materials and Methods" since different data sources are described.

Ans: Agreed, revised accordingly.

4) g., in L. 271, there is a double bracket in the manuscript 'Bertoldi et al. (2010))'. There are more of these in the manuscript, which should be avoided.

Ans: This is due to one bracket closing the year of the citation, while the other closes the textual statement, so albeit not visually pleasing, it is correct according to the journal's citation style. We checked all occurrences of double closing brackets and they should all fall in this category.

**Appendix A: Further considerations on the definition of hydropower variables**

In this appendix, we provide additional insights into the definition of key hydropower variables, specifically $Q_{AVG}$, $Q_{DES}$, and $Q_{MAX}$, and discuss their mutual relationships. The flow variables are defined as it follows:

600
$$Q_{AVG} = \frac{E_{AVG}}{\gamma H \eta \times 24 \times 365} \tag{A1}$$

$$Q_{DES} = \frac{W_{inst}}{\gamma H \eta} \tag{A2}$$

all terms are defined in Table 2. $Q_{AVG}$ thus represents the average turbined flow assuming the plant is functioning all hours throughout the year, so it is representative in terms of aggregated turbined volumes, less so in terms of daily turbine operation
605 (i.e., it does not represent the value *at which* the plant usually operates, but rather the average of the functioning and non-functioning hours throughout the year. $Q_{DES}$ represents the design flow rate of a given hydropower system and is computed based on its installed capacity: it therefore represents the flow rate at which the plant achieves its optimal power output, $W_{ints}$. Finally, $Q_{MAX}$ represents the maximum authorized discharge rate for both water intakes and reservoirs (named $Q_{work}$ for the latter, see Table 2). This means that it represents a regulatory constraint, rather than a hydraulic characteristic of the system.
610 Figure A1 depicts the mutual relationship between $Q_{AVG}$, $Q_{DES}$, and $Q_{MAX}$ for all reservoir hydropower systems, sorted according to $Q_{MAX}$.

As it can be expected, $Q_{AVG}$ is consistently lower, at varying rates, than the other two (average ratio $Q_{AVG}/Q_{MAX} = 0.33$, hereafter referred to as Q/Q ratio). Interestingly, $Q_{DES}$ and $Q_{MAX}$ appear very similar through all systems, while one might expect $Q_{DES}$ to be somewhat lower. The explanation lies in the fact that $Q_{MAX}$ is not a hydraulic capacity value, but rather
615 a regulatory one. Indeed, hydropower systems are often designed after their respective maximum authorized flow rate, for the sake of cost-efficiency. This tells us that the recorded value of $Q_{MAX}$ can reasonably be assumed as the design flow rate for each hydropower system.

We now investigate the relationship between the Q/Q ratio and other system properties such as head, active volume, and regulation capacity (Rc).
620 The active volume is defined as the volume available for regulation activities, between $H_{min,reg}$ and $H_{max,reg}$, and is obtained interpolating the respective stage-storage curve for each reservoir. Rc is defined as the time (days) needed to fill the active volume with the average inflow. In the absence of official information, the average inflow to reservoirs was modeled under natural conditions for the 1995-2008 time window, following the same setup described in Section 2.2.

The relationship between the Q/Q ratio and the other system characteristics is summarized in Figure A2: the upper three
625 panels (a-c) show the univariate correlation between Q/Q and system head, active volume and regulation capacity, respectively. A linear regression analysis highlights that both head and regulation capacity have a significant, inverse proportionality with Q/Q. On the other hand, the active volume exhibits no significant correlation. The results of the linear regression analysis are summarized in Table A1. The interplay between these variables in shaping the Q/Q ratio for each hydropower system is

[Figure]

**Figure A1.** Characteristic discharge values for each reservoir hydropower system. Systems are sorted along the x-axis in order of increasing $Q_{MAX}$, for readability.

depicted in Figure A2d: a rather defined trend appears, confirming that hydropower systems with low regulation capacity (red bubbles) often have a higher Q/Q ratio, meaning they most times turbine close to their design capacity. This is reasonable, since this category of reservoirs tends to fill up more quickly, as opposed to the ones with high regulation capacity, which are often operated in peaking mode to exploit the maximum available head and flow (likely during periods of high power demand), leveraging their regulation capacity and at the same time resulting in significant down time. This behavior is reinforced by head, as hydropower systems with low head and regulation capacity exhibit the highest Q/Q ratios. However, we see no clear operational explanation for the relationship between system head and its Q/Q ratio, unless low head is simply a proxy of reservoirs systems designed to operate in almost continuous conditions (i.e., Run-of-the-river-like, whence with low regulation capacity). Finally, volume alone seems, as also highlighted in the univariate linear regression, to bear no relevance towards the resulting Q/Q ratio: volume has no relationship with system head (hydropower systems with different storage also have

different heads with no correlation) nor with regulation capacity (as it depends on the inflow, which doesn't strictly vary ccording to reservoir volume).

**Table A1.** Goodness-of-fit statistics for the linear regression models fitted between Q/Q Ratio and the reservoir systems' head, active volume and regulation capacity.

| Q/Q Ratio Linear Model | Head | Active Volume | Regulation Capacity |
|---|---|---|---|
| R-squared | 0.1385 | 0.00237 | 0.1852 |
| p value | $1.12 \times 10^{-5}$ | 0.5793 | $2.59 \times 10^{-7}$ |

We finally conducted a k-means clustering analysis to verify our hypotheses on the relationship between Q/Q ratio and system characteristics. There were no clear (analytical) indications of an optimal number of clusters: our attempts highlighted $n = 3$ clusters as a good candidate, while the Bayesian Information Criterion indicates $n = 6$ as optimal. Hence, we performed the clustering for both values of $n$, but noticed that $n = 6$ produced unstable results due to the heterogeneity of the systems' characteristics (clusters with 1 and 5 members, and no additional information compared to $n = 3$). Thus, we stuck to $n = 3$: we summarized the centroids' coordinates in Table A2 and we visualized them in Figure A2e: we see two clear, opposite clusters: systems with medium-high regulation capacity (cluster 3, gathering most black bubbles from FigureA2d) and low Q/Q ratio, and systems with low regulation capacity and correspondingly higher Q/Q ratio (cluster 1, gathering red bubbles from the previous panel). Finally, a cluster emerges for few systems with very high regulation capacity (> 800 days, see cluster 2 and corresponding yellow bubbles in the previous panel): these are characterized by medium-sized reservoirs, of which the larger tend to also have higher Q/Q ratio, possibly because they were designed to accommodate large incoming flows and are operated accordingly. All things considered, this analysis showed that regulation capacity is a good, yet not exhaustive, proxy of the typical reservoir operation regime of each system, here synthesized by the Q/Q ratio. The exact relationship between the operation of individual systems and their structural characteristics is far more complex, and accounts for environmental and managerial aspects that are impossible to capture at this scale. Finally, we would like to remark that the regulation capacity is computed based on modeled inflows and, as such, is prone to local errors (for instance, we do believe that the bright yellow bubble in Figure A2d is originated by an unreasonably low modeled inflow). Thus, we conclude by stressing the importance of thoroughly assessing the hydrological characteristics of each catchment to better understand how they influence the operation of the hydropower systems therein.

**Table A2.** Centroid Coordinates from k-means clustering ($n = 3$).

| Cluster | QQ | Head [m] | Active volume [Mm$^3$] | Regulation capacity [days] |
|---|---|---|---|---|
| 1 | 0.500 | 246 | 9.56 | 26.3 |
| 2 | 0.236 | 434 | 121.0 | 1555 |
| 3 | 0.242 | 641 | 18.4 | 291 |

[Figure]

**Figure A2.** Relationship of Q/Q ratio with structural characteristics of the related hydropower system. Panels (a-c) show the univariate relationship with system head, reservoir active volume, and regulation capacity, respectively, together with a linear regression model (black line) and the corresponding 95% confidence interval (gray shading). Panel (d) shows the mutual relationship of all four variables, using color and size of the bubbles to track their regulation capacity and active volume, respectively, while head was assigned to the x-axis to improve readability. Panel (e) shows the clusters resulting from the k-means clustering analysis, using 3 clusters. The bubble sizing was left unchanged from the previous panel, to allow easier identification of the individual systems.

**Supporting Information**

**Comprehensive inventory of large hydropower systems in the Italian Alpine Region**

Andrea Galletti, Soroush Zarghami Dastjerdi, Bruno Majone

**1. Database Format**

**1.1 Infrastructures Location**

Coordination of human systems infrastructures and their connections provided in two divided shapefiles (.shp) consisted of:
Channel/Penstock.shp
IAR-HP.shp

A shapefile representing the Italian Apline Region: Italian_Alpine_Region.shp

All Shapefiles are all provided on the WGS84/UTM zone 32N projection (EPSG: 32632), and are stored in the shapefiles.zip archive.

**1.2 IAR-HP**

The comprehensive attributes for each hydropower system are provided in an Excel file (.xlsx):

IAR-HP.xlsx

The file contains seven sheets as following and the variables included in each column of these sheet are as explained in the following tables.

**1.2.1   Nomenclature**

A general explanation of each characteristic abbreviation inside the dataset and its description and unit.

**1.2.2  IAR-HP**

Sheet containing a comprehensive guideline on the hydropower systems topology and their geographical information.

| IAR-HP | | | |
|---|---|---|---|
| **Parameter** | **Notation** | **Description** | **Columns** |
| Representative ID of the infrastructure | ID_NODE | - | 1 |
| Infrastructure Longitude | X_Coord | - | 2 |
| Infrastructure Latitude | Y_Coord | - | 3 |
| Infrastructure Altitude | Z_Coord | - | 4 |
| Infrastructure name | Name | - | 5 |
| Type of the infrastructure | Type | Divided into three group of (Intake, Reservoir, Plant), can be filtered via their initials | 6 |
| Basin | Basin | - | 7 |
| Province | Province | - | 8 |
| Region | Region | - | 9 |
| Numerator of Downstream nodes | $\#link_{Down}$ | Number of Downstream nodes | 10 |
| Downstream node ID | $ID_{DOWN}$ | Representative ID of Downstream node | 11 |
| Numerator of Upstream nodes | $\#link_{UP}$ | Number of Upstream nodes | 12 |
| Upstream node ID | $ID_{UP}$ | Representative ID of Upstream node, in case of having a node receiving water from more than one upstream, column 14 shows the ID of the second upstream node | 13-14 |
| Numerator of the plant references | $\#plt_{ref}$ | Number of Plant references | 15 |
| Plant references ID | $ID_{ref}$ | Representative ID of the plant references, in case of having a plant having more than one reference, column 17 shows the ID of the second reference | 16-17 |

**1.2.3 RES_data**

Sheet detailing the infrastructural characteristics of reservoirs.

| Parameter | Notation | Units | Description | Columns |
|---|---|---|---|---|
| Spillway crest elevation | $H_{max,inv}$ | m.a.s.l | - | 11 |
| Maximum regulation stage | $H_{max,reg}$ | m.a.s.l | Highest regulated level at which water can be stored in a reservoir without spilling | 12 |
| Minimum regulation stage | $H_{min,reg}$ | m.a.s.l | Lowest regulated level in a reservoir allowed to be used for hydropower purposes | 13 |
| Maximum authorized flow rate threshold | $H_{work}$ | m.a.s.l | Reservoir level threshold at 90% of available stage range, beyond which water is turbined at maximum authorized flow rate to prevent spilling and ensure operational safety | 14 |
| Maximum authorized turbine flow rate | $Q_{work}$ | $m^3/s$ | Maximum authorized turbine flow rate | 15 |
| Spillway flow rate | $Q_{spill}$ | $m^3/s$ | - | 16 |
| Starting storage level | $H_{zero}$ | $m^3/s$ | Boundary condition for modelling purposes | 17 |
| Gross system head | $H_{AVG}$ | m | Nominal average head reported for the reservoir | 18 |
| Long-term average nominal production | $E_{AVG}$ | Gwh/year | - | 19 |
| Long-term average turbined flow | $Q_{AVG}$ | $m^3/s$ | Long-term average turbined flow based on long-term average nominal production assuming the plant has operated full time | 20 |
| Number of discretizations of the stage storage curve | ndisc | Integer | Value written in the cell represents as pair of discretizations as V(H) | 21 |
| Stage-Storage curve | V(H) | $H_n$= m.a.s.l $V_n$= $Mm^3$ | Odd columns are the level discretization followed by the even columns showing the equivalent volume at that level | 22 – Nth column |

*First ten columns of this sheet contain the topological and geographical information explained in IAR-HP sheet

**1.2.4 PLT_data**

Sheet detailing the infrastructural characteristics of hydropower plants.

| PLT-data* | | | | |
|---|---|---|---|---|
| **Parameter** | **Notation** | **Units** | **Description** | **Columns** |
| Hydropower plant type | Subtype | - | Hydropower plants are classified into two main categories: Reservoir-based Hydropower (R) and Run-of-the-River (RoR). If a plant receives water from multiple sources for power generation, its hydropower type is specified in column 16 | 15-16 |
| Hydropower head | H | m | Height differences with the plant references | 17-18 |
| Pumped-storage hydropower plant | PSH | - | Hydropower plants are further categorized into three subgroups: Pure Pumped Storage (PSH), Mixed Pumped Storage (M-PSH), and Conventional (None) in case of absence of any type of pumping | 19 |
| Installed power | $W_{inst}$ | Mw | Nameplate capacity, Maximum achievable power output under optimal operating conditions | 20 |

*First fourteen columns of this sheet contain the topological and geographical information explained in IAR-HP sheet

**1.2.5 ITK_data**

Sheet detailing the infrastructural characteristics of water intakes (channels/penstocks).

| ITK-data* | | | | |
|---|---|---|---|---|
| **Parameter** | **Notation** | **Units** | **Description** | **Columns** |
| Name | - | - | Suffixes at the end of each naming identifies the intakes purposes as: RoR Intakes (_I# suffix), Subsequent Intakes (_OP# suffix), Confluence (_B# suffix) | - |
| Maximum authorized flow intake | $Q_{MAX}$ | $m^3/s$ | - | 15 |

*First fourteen columns of this sheet contain the topological and geographical information explained in IAR-HP sheet

**1.2.6 Q_RULE**

Sheet containing the expected turbine flow rate for reservoirs' operational reference.

| Q_RULE* | | | | |
|---|---|---|---|---|
| **Parameter** | **Notation** | **Units** | **Description** | **Columns** |
| Long-term average turbined flow | $Q_{AVG}$ | m³/s | See RES_data | 4 |
| Expected monthly turbine rate | Monthly data (Jan–Dec) | m³/s | Product of $Q_{AVG}$ with the monthly turbine coefficient | 5-16 |
| Monthly turbine rate coefficient | Monthly data (Jan–Dec) | - | Ratio between long-term monthly hydropower production and long-term annual hydropower production. Computed for each month and each province | 19-30 |

*First three columns of this sheet contain the topological and geographical information explained in IAR-HP sheet

**1.2.7 Q_MEF**

Sheet specifying the minimum ecological flow requirements for each infrastructure.

| Q_MEF* | | | | |
|---|---|---|---|---|
| **Parameter** | **Notation** | **Units** | **Description** | **Columns** |
| Minimum ecological flow | $Q_{MEF}$ Monthly data (Jan–Dec) | m³/s | Minimum monthly environmental flow threshold below which no hydropower diversions can occur. For modelling purposes, in the specific case of confluence intakes, which do not divert additional water, a symbolic value of 999 m³/s is adopted. | 4-15 |

*First three columns of this sheet contain the topological and geographical information explained in IAR-HP sheet

**2. IAR-HP Reassembly Guidance**

The Excel file is designed to allow anyone to reconstruct the entire layout shown in Figure 1 of the paper (Galletti et al, 2025), or to write the topology for modelling purposes, without any prior knowledge of the systems' layout. Any topology-related addressing (i.e., the ID's shown as IDUP and IDDOWN for every infrastructure) refer to the IDs in the IAR-HP sheet, which are just copy-pasted into the subsequent sheets for reference.

For building a topology, the user should solely consider the IDs on the IAR-HP sheet, regardless of the structure types involved (ITK, RES, PLT) in their respective sheets. Figure 1 presents an overview of the basic human systems layouts and their interconnections and how they are introduced in the IAR-HP dataset.

[Figure]

*Figure 1 An overview of the different human systems infrastructures' interconnection in IAR-HP*

Let us construct a fairly complex topology step by step. Let's say we are interested in the system feeding the second plant in IAR-HP. First of all, we move directly to IAR-HP and stick with it. The first thing we notice is that the ID of said plant is **ID 13** (Cogolo) shown in Figure2:

[Figure]

*Figure 2. Cogolo hydropower system topology*

- **ID 13:** The upstream IDs (ID$_{UP}$) are **ID4** and **ID12**, two intakes (this makes no difference in constructing the topology, but let us keep track of node types for when we finally assemble the topology). There is no downstream ID, because it is a plant and our model assumes that all plants discharge in the river[1].

Now, let us look upstream, looking at **ID4** first, which originates the first "branch" of our system.

- **ID 4:** The downstream node is **ID13** (the one we just investigated) and the upstream node is **ID3**. Once again, an intake. Let us then continue checking upstream nodes.
* * *
[1] If a plant discharges in a channel feeding a downstream plant instead, this is taken care of by the presence of an ITK immediately downstream of the plant outlet, so water does not return to the natural network.

- **ID 3:** The downstream node is **ID4** (correct) and upstream is **ID2**, another intake.
- **ID 2:** Downstream is **ID3** and the upstream is **ID1**, a reservoir.
- **ID 1:** Downstream is **ID2**, and there is no upstream ID[2]. We can therefore stop looking upstream for this branch.

So, on this side we have a sequence of R(1)-I(2)-I(3)-I(4)-P(13).

Let us now look at the second branch, starting from ID12, more briefly:

- **ID 12,** an intake, has **ID13** downstream (the plant) and **ID11** upstream, another intake.
- **ID 11** is an intake, we confirm ID12 downstream, while **ID 10** is upstream (intake).
- **ID 10** (intake) has **ID11** downstream and **ID 9**, upstream, a reservoir.
- **ID 9** has no upstream nodes. This completes this branch

On this side we have a sequence of R(9)-I(10)-I(11)-I(12)-P(13).

Thus, on both sides there are two reservoirs (IDs 1 and 9) each supplying a channel with three subsequent intakes. These two channels meet at ID 13, which is a plant that considers the two channels separately[3]. The resulting system is sketched in Figure 3.
* * *
[2] Reservoirs often have no connected structures upstream, as they capture the entire underlying drainage basin; the common occurrence of water diversions from other catchments is handled by placing the outlet of a water diversion channel hydrologically upstream of the reservoir, so that it can collect water from it naturally.

[3] Meaning each branch produces hydropower based on its respective load difference. This is the case when the column #plt_ref has a number greater than one, meaning that there is more than one location that should be considered when computing the head difference between the water intake and the turbine, as opposed to the default case of #plt_ref =1 where the plant considers the head difference with only one upstream node. In other cases, a confluence between multiple might happen in a surcharge basin before the plant. Confluences are highlighted by suffix "_B" in the dataset.

[Figure]

*Figure 3 Cogolo hydropower system resemblance trace, illustrating the system's structure and connections as referenced in the tutorial*

**3. Hydropower Plants References Catchment Boundaries**

A ZIP file contains the catchment areas of all water inflow sources for hydropower systems as GeoTIFF (.tiff) files. Each file is named according to the ID of the corresponding infrastructure in the dataset. The pyGIS code adopted in QGIS to automate the delineation of the catchments draining into each infrastructure is provided below. Required inputs are (an Excel file containing the ID_node of the references and their coordination and the drainage directions file), and the produced output files are named according to the ID of the corresponding infrastructure in the dataset, following the format:

out_#.tiff (where # represents the infrastructure ID). GeoTIFF files are provided on the WGS84/UTM zone 32N projection (EPSG: 32632), and are stored in the Plant references TIFF.zip archive.

**3.1 Hydropower Plants References Catchment Boundaries pyGIS code**

```python
import processing
import pandas as pd

**Define file paths**
id_file = ''  # File with columns: id, x, y of the references
input_water_file = '/dd.csv' # Drainage directions file
output_base = '/out'  # Base path for output files

**Read the Excel file with outlets' coordinates**
nodes_df = pd.read_excel(id_file)

**Loop through each node and run the water outlet calculation**
for index, row in nodes_df.iterrows():
    node_id = row['id']
    x = row['x']
    y = row['y']

    # Create the coordinate string in the required format (e.g., "x,y [EPSG:32632]")
    coord = f"{x},{y} [EPSG:32632]"

    # Construct the output filename; e.g., "out_7.tiff" for node with id 7
    output_file = f"{output_base}_{node_id}.tiff"

    # Run GRASS r.water.outlet
    processing.run("grass7:r.water.outlet", {
        'input': input_water_file,
        'coordinates': coord,
        'output': output_file,
        'GRASS_REGION_PARAMETER': None,
        'GRASS_REGION_CELLSIZE_PARAMETER': 0,
        'GRASS_RASTER_FORMAT_OPT': '',
        'GRASS_RASTER_FORMAT_META': ''
    })

    print(f"Processed node {node_id} at ({x}, {y}); output saved to {output_file}")
```

**4. References**

Galletti, A., Zarghami Dastjerdi, S., and Majone, B.: Comprehensive inventory of large hydropower systems in the Italian Alpine Region, Earth Syst. Sci. Data Discuss. [preprint], https://doi.org/10.5194/essd-2024-521, in review, 2025.